# Cluster-based characterization of multi-dimensional tropospheric ozone variability in coastal regions: an analysis of lidar measurements and model results

Claudia Bernier[1], Yuxuan Wang[1], Guillaume Gronoff[2,3], Timothy Berkoff[2], K. Emma Knowland[4,5], John T. Sullivan[4], Ruben Delgado[6,7], Vanessa Caicedo[6,7], Brian Carroll[2,6]

[1] Department of Earth and Atmospheric Science, University of Houston, Houston, Texas, USA

[2] NASA Langley Research Center, Hampton, VA, 23666, USA

[3] Science Systems and Application Inc., Hampton, VA, 23666, USA

[4] NASA Goddard Space Flight Center, Global Modeling and Assimilation Office, Greenbelt, MD, 20771, USA

[5] Morgan State University, Goddard Earth Science Technology & Research (GESTAR) II, Baltimore, Maryland, USA

[6] Joint Center for Earth Systems Technology, Baltimore, MD, USA

[7] University of Maryland Baltimore County, Baltimore, MD, USA

*Correspondence*: Yuxuan Wang (ywang246@central.uh.edu)

**Abstract.** Coastal regions are susceptible to multiple complex dynamic and chemical mechanisms and emission sources that lead to frequently observed large tropospheric ozone variations. These large ozone variations occur on a meso-scale which have proven to be arduous to simulate using chemical transport models (CTMs). We present a clustering analysis of multi-dimensional measurements from ozone Light Detection And Ranging (LiDAR) in conjunction with both an offline GEOS-Chem CTM simulation and the online GEOS-Chem simulation GEOS-CF, to investigate the vertical and temporal variability of coastal ozone during three recent air quality campaigns: 2017 Ozone Water-Land Environmental Transition Study (OWLETS)-1, 2018 OWLETS-2, and 2018 Long Island Sound Tropospheric Ozone Study (LISTOS). We developed and tested a clustering method that resulted in 5 ozone profile curtain clusters. The established 5 clusters all varied significantly in ozone magnitude vertically and temporally which allowed us to characterize the coastal ozone behavior. The lidar clusters provided a simplified way to evaluate the two CTMs for their performance of diverse coastal ozone cases. An overall evaluation of the models reveals good agreement ($R \approx 0.70$) in the low-level altitude range (0 to 2000 m), with a low and unsystematic bias for GEOS-Chem and high systemic positive bias for GEOS-CF. The mid-level (2000 – 4000 m) performances show a high systematic negative bias for GEOS-Chem and an overall low unsystematic bias for GEOS-CF and a generally weak agreement to the lidar observations ($R = 0.12$ and $0.22$, respectively). In evaluating the cluster specific performances additional model insight is revealed as cluster-by-cluster model performance is more convoluted than the overall performances suggest. Utilizing the full vertical and diurnal ozone distribution information specific to lidar measurements, this work provides new insights on model's proficiency in complex coastal regions.

## 1. Introduction

Tropospheric ozone ($O_3$) is an important secondary pollutant created by multiple reactions involving sunlight, nitrogen oxides ($NO_x = NO + NO_2$), and volatile organic compounds (VOCs) which, in accumulation, can have damaging effects on human and plant health. In addition to its photochemical growth, $O_3$ can easily be influenced by local and regional transport mechanisms. For coastal regions, surface $O_3$ is highly variable in time and space due to its susceptibility to many factors such as local ship emissions, long range transport, and sea/bay breeze processes. Multiple studies have proven the strong influence that sea/bay breeze and wind flow patterns can have on the accumulation of coastal $O_3$ and can often lead to poor air quality (e.g., Tucker et al., 2010; Martins et al., 2012; Stauffer et al., 2012; Li et al., 2020). Loughner et al. (2014) highlighted the importance for understanding the ability for bay breeze events to cause $O_3$ differences not only spatially but vertically in coastal regions.

This variability is challenging for air quality models to capture as high-resolution measurements are necessary to fully understand and simulate this $O_3$ behavior in coastal regions. For example, Dreessen et al. (2019) tested the U.S. Environmental Protection Agency (EPA) Community Multiscale Air Quality (CMAQ) model's ability, configured at 12 km, to simulate $O_3$ exceedances at Hart Miller Island in Maryland (HMI) revealing high bias and 'false alarms' due to several reasons such as emission transport over water and the coarse model resolution's inability to capture fine-scale meteorology and transport. Cases such as sea/bay breeze events, which directly contribute to high coastal $O_3$ cases, are denoted by local meteorological mechanisms such as surface wind speed deceleration, wind direction convergence and recirculation (Banta et al., 2005). Air quality models with coarse horizontal and vertical resolutions are not able to capture such fine developments (Caicedo et al., 2019). Ring et al. (2018) also used CMAQ to estimate the impact of ship emissions on the air quality in eastern U.S. coastal regions indicating that an understanding of the vertical profiles of emissions was significant for improving air quality simulations. These are consistent and unanimous issues with air quality modeling in coastal regions. Since offshore sites within coastal regions are historically under sampled due to the difficulty of water-based measurements, this problem is still pertinent today.

Recently, three associated air quality campaigns set out to address this issue (https://www-air.larc.nasa.gov/index.html): 2017 & 2018 NASA Ozone Water-Land Environmental Transition Study (OWLETS-1 & OWLETS-2) and Long Island Sound Tropospheric Ozone Study (LISTOS) (e.g., Sullivan et al., 2019). These three campaigns were each conducted in highly populated coastal regions along the Chesapeake Bay in Virginia and Maryland and Long Island Sound in the New England/Middle Atlantic region, respectively, that are vulnerable to $O_3$ exceedances with the goal of filling the measurement gaps in these regions. During these campaigns, a suite of detailed airborne and ground measurements were taken during the course of highly polluted summer months (end of May through August) to capture the variability of pollutants, including $O_3$ and its precursor species, and the distinct meteorological processes specific to land-water regions that affect them.

The three campaigns strategically placed multi-dimensional tropospheric lidar measurements of $O_3$ on and offshore in order to capture critical land-water gradients and to fill the deficit of measurements in these under monitored areas. These measurements were supported as part of NASA's Tropospheric Ozone Lidar Network (TOLNet). Continuous profile

measurements from O₃ lidars highlight important regional transport and temporal variations of $O_3$ in the lower and middle
levels of the troposphere that are usually difficult to capture by most satellite-based remote-sensing instruments (Thompson et
al., 2014). Lidar measurements are unique in their ability to capture high resolution full $O_3$ 2-D profile curtains over a period
of time that indicate pollutant transport and can help in understanding $O_3$ behavior in coastal regions. In Gronoff et al. (2019),
the co-located lidar at the Chesapeake Bay Tunnel Bridge (CBBT) during OWLETS-1 successfully captured a near-surface
maritime ship plume emission event on August 01, 2017. An ensemble of other instruments (e.g., drones, Pandora spectrometer
systems, etc.) launched near the shipping channel captured elevated $NO_2$ concentrations while the lidar instrument captured a
depletion of $O_3$ simultaneously. The lidar was able to capture the unique low range altitude $O_3$ concentrations which elucidated
the evolution of the trace-gas concentrations during this ship plume event.
Several studies have thoroughly evaluated the results from the air quality campaigns used in this study but were focused
more on specific case studies (Dacic et al., 2019; Sullivan et al., 2019; Gronoff et al., 2019). Dacic et al. (2019) used lidar
measurements of a high $O_3$ episode during OWLETS-1 to evaluate the ability of two NASA coupled chemistry-meteorology
models (CCMMs), the GEOS Composition Forecast ("GEOS-CF"; Keller et al., 2021) and MERRA2-GMI (Strode et al.,
2019), to simulate this high $O_3$ event. They found that the GEOS-CF model performed fairly in simulating $O_3$ in the lower
level (between 400 to 2000 m ASL) and outperformed MERRA2-GMI based on surface observations at multiple monitoring
sites and by a median difference of -6 to 8 % +/- 7 % at both lidar sites. In the case of this event, GEOS-CF was able to simulate
the 2-D $O_3$ profile curtains at small scales. At the time of the Dacic et al. (2019) study, only processed observational data from
OWLETS-1 was available.
For this study, we took advantage of 91 captured 2-D (vertical and diurnal) $O_3$ profile curtains from all three air quality
campaigns (Sect. 2). To characterize the different behaviors of $O_3$ in coastal regions, we developed a novel clustering method
based on the altitude and time dimensions of the lidar measurements that organized the profile curtains (Sect. 2). We used the
developed clusters to evaluate the ability of both offline and online GEOS-Chem and GEOS-CF simulations to reproduce the
coastal $O_3$ and wind characteristics highlighted by each cluster (Sect. 3).

**2. Materials & Method**
**2.1. Air quality campaigns**
During the years 2017 and 2018, NASA in partnership with other U.S. national agencies and university research groups
orchestrated three air quality campaign studies that focused on key land and water observations: OWLETS-1, OWLETS-2,
and LISTOS. OWLETS-1 was conducted in 2017 from July 5 to August 3, while OWLETS-2 and LISTOS were conducted in
2018 from June 6 to July 6 and July 12 to August 29, respectively. All campaigns took advantage of a multitude of ground,
aircraft, and remote sensing measurements. For the sake of this study, we will focus on measurements from the two lidars from
the TOLNet: NASA Langley Mobile Ozone Lidar (LMOL) (De Young et al. 2017; Farris et al. 2018; Gronoff et al, 2019,
2021) and NASA Goddard Space Flight Center (GSFC) Tropospheric Ozone (TROPOZ) Differential Absorption Lidar (DIAL)
(Sullivan et al. 2014, 2015a), which ran simultaneously at the marked positions in Figure 1. The TOLNet data from all three
campaigns are available on the NASA LaRC Airborne Science Data for Atmospheric Composition archive (https://www-
air.larc.nasa.gov/missions.htm; accessed – 20 January 2021).

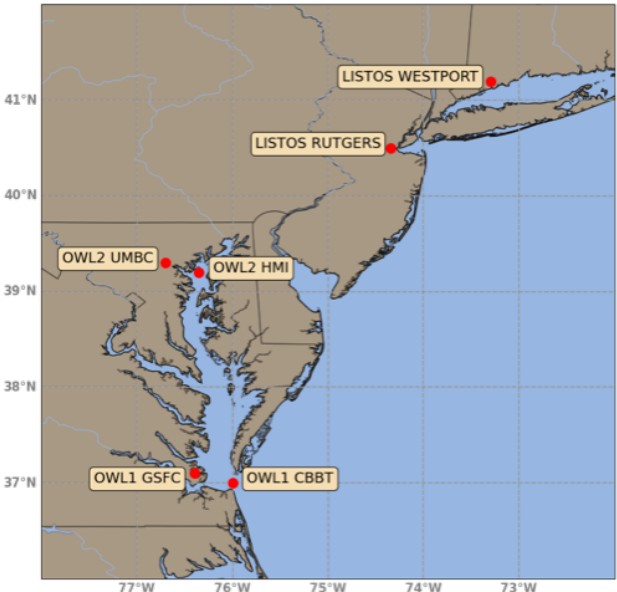

**Figure 1.** An inset map of the Chesapeake Bay airshed in Maryland, Virginia, and Long Island Sound in New York with the
six lidar monitoring locations used for OWLETS-1, OWLETS-2, and LISTOS highlighted and labeled.

The two lidars were placed strategically for each campaign (Figure 1), so that one lidar was closest to over-water
measurements while the other was farther inland with the goal of examining how $O_3$ transport and concentration is influenced
by specific coastal mechanisms such as the land–water breezes. For OWLETS-1, the LMOL lidar was used at the CBBT
[37.0366°N, 76.0767°W], depicting the real time over water $O_3$ measurements while the GSFC TROPOZ lidar was stationed
at NASA Langley Center [37.1024°N, 76.3929°W] further inland. Similarly, for OWLETS-2, the LMOL lidar was stationed
for the over water measurements at Hart Miller Island [39.2449° N, 76.3583° W] and GSFC TROPOZ was stationed at the
University of Maryland, Baltimore County (UMBC) [39.2557° N, 76.7111° W]. Finally, for LISTOS, LMOL was at the
Westport site [41.1415° N, 73.3579° W] and TROPOZ at Rutgers [40.2823° N, 74.2525° W]. For the sake of this study the
unique benefits due to the different placements (onshore versus offshore) of the co-located lidars are not specifically evaluated.
Instead, the study focuses on the benefits of detailed and multi-dimensionality of both lidar instrument data in general.

Routine lidar measurements were taken for the duration of the campaigns providing 91 multi-dimensional $O_3$ profile
curtains. Both lidars retrieve data at a 5-min temporal resolution and use a common processing scheme to produce a final $O_3$
product which was used for this study. In this study, the individual profile curtains refer to the "full day", vertical and diurnal

lidar measurements. In this study, 91 individual 2-D profile curtains were used from both lidars from the three campaigns: 26
profile curtains from OWLETS-1, 28 profile curtains from OWLETS-2, and 37 profile curtains from LISTOS.
To evaluate meteorological impacts on the lidar $O_3$ clusters and distinguish certain model discrepancies we used various
temperature and wind measurements. Hourly observed temperature, wind speed and wind direction, and $O_3$ from surface
monitors pertaining to the study area were obtained from the Air Quality System (AQS) (data can be accessed at
https://aqs.epa.gov/aqsweb/airdata/). Along with the $O_3$ lidar instruments, we utilized high resolution vertical and horizontal
wind speed and direction data monitored by Doppler wind lidar Leosphere WINDCUBE 200s instruments deployed at HMI
during OWLETS-2 during LISTOS (e.g., Couillard et al., 2021; Coggon et al., 2021; Wu et al., 2021).

**2.2**. **Clustering lidar data**
**2.2.1 Description of the ozone lidar measurements**
The lidar instrument is unique in that it provides high dimensional profile measurements of $O_3$, as opposed to one
dimensional surface measurements from air quality monitoring sites. The two TOLNet lidars used during the campaigns have
been evaluated for their accuracy during previous air quality campaigns (DISCOVER-AQ; https://www-
air.larc.nasa.gov/missions/discover-aq and FRAPPÉ; https://www2.acom.ucar.edu/frappe) and have also been compared
against each other (e.g., Sullivan et al., 2015; Wang et al., 2017). The two lidars have different transmitter and retrieval
components but produce $O_3$ profiles within 10 % of each other as well as compared to ozonesondes (Sullivan et al., 2015). In
comparison with other in situ instrument measurements, the TOLNet lidars were found to have an accuracy better than ±15 %
for capturing high temporal tropospheric $O_3$ vertically proving their capability of capturing high temporal tropospheric $O_3$
variability (Wang et al., 2017; Leblanc et al., 2018).
To characterize coastal $O_3$ during the summer months, we use a multitude of lidar profile curtains obtained during the
OWLETS-1, 2, and LISTOS campaigns. The two lidars used in the campaigns produced profile curtains of $O_3$ from 0 – 6000
m above ground level (AGL) with some days beginning as early as 06:00 local time (EDT) and ending measurements as late
as the last hour of the day. One of the challenges is that the multiple lidar datasets are not always uniform; although most of
the profile curtains began at or around 08:00 EDT, the lidar measurements commence and conclude at different times. At the
time of these campaigns, the lidar data retrieval was constrained by the availability of personnel as well as the availability of
electricity in remote areas (at time of writing, the lidar instrument systems have been updated and are now more fully
automatized for use during succeeding campaigns removing such constraints). Due to this constraint, the 91 lidar curtains
range from as short as a 6-hour window to a full 24-hour window. Similarly, the profile curtains do not have an exact uniform
altitude range either. In the processing of the lidar data, some measurements may be filtered out and removed due to issues,
such as clouds, which can influence and degrade the retrieval leaving some blocks of empty data within the vertical altitude
dimension. When the cloud conditions are perfect, the limiting factor for the altitude is the solar background: the UV from the
sun is a source of noise that prevents the detection of the low level of backscattered photons. For LMOL, this means that the
maximum altitude is about 10 km AGL at night (Gronoff et al., 2021) and lowered to about 4 km AGL at solar noon (worse
conditions possible for the summer in the continental U.S. resulting in below 4 km AGL). This results in a general scarcity of
$O_3$ measurements above 4000 m AGL for most of the vertical profile curtains. Lidars still have limitations that prove to be a
complication e.g., noise signal and manual operations. At the time of writing, the operative limitation has been addressed and
the lidars are now more fully automated which removes some of the difficulty.

**2.2.2 Clustering approach and application**
To facilitate the comparison of the 2-D $O_3$ profile curtains and the air quality model simulations we used a cluster analysis
that categorized the behavior of the tropospheric $O_3$ captured in the profile curtains. Clustering methods are commonly used
in air quality and atmospheric studies to group and characterize large datasets (Darby, 2005; Alonso et al., 2006; Christiansen,
2007; Davis et al., 2010; Stauffer et al., 2018). In our previous work, we have successfully used clustering methods to
automatically characterize diurnal patterns of surface winds and surface $O_3$ in the Houston-Galveston-Brazoria area that proved
to perform better than a rudimentary quantile method to reveal the dependence of surface $O_3$ variability on local and synoptic
circulation patterns on the Gulf Coast (Bernier et al., 2019; Li et al., 2020)
In evaluating the structure of the lidar measurements and working within measurement limitations (described in Sect.
2.2.1) from the three air quality campaigns, we developed a method to cluster multi-dimensional $O_3$ profile curtains using K-
Means clustering algorithm. Input features (seed values) were rationally established to best represent the behavior of $O_3$
temporally and vertically without including an excessive amount of input features, which can weaken the results of clustering
(discussed in detail in Sect. S1). With the goal of evaluating lower level tropospheric $O_3$ and based on description of the
structure and constraints of the lidar measurements, the features were tailored to the altitude range 0 – 4000 m AGL and time
range of 06:00 EDT – 21:00 EDT.
Figure 2 illustrates the 8 features that represent slabs of altitude and time used in the cluster analysis. For each $O_3$ profile
curtain (total of 91), we calculated the average $O_3$ from the following time and altitude range: Features 1 – 4 altitudes range
from 0 – 2000 m; Features 5 – 8 altitudes range from 2000 – 4000 m. The two altitude ranges were determined to best represent
different $O_3$ transport events although they do not explicitly represent these layers. For Features 1 – 4, $O_3$ would most likely
primarily be affected by local production and pollution transport while for Features 5 – 8, $O_3$ would more likely be associated
with long range transport (e.g. interstate). As planetary boundary layer growth (PBL) in coastal regions do not usually reach
altitudes greater than 2000 m, mixing between the boundary layer and free troposphere would presumably take place within
the low-level altitude bin. Additional attention to the PBL in the selecting of low versus mid-level features for the clustering
will be investigated in future work. For clarity, we will use the terms low-level and mid-level features to address the two
altitude subsets e.g., Features 1 – 4 and 5 – 8, respectively. Feature 1 and 5 time range from 06:00 – 08:00 EDT; Feature 2 and
6 from 08:00 – 12:00 EDT; Feature 3 and 7 from 12:00 – 16:00 EDT; and Feature 4 and 8 from 16:00 – 21:00 EDT. The four
subset time ranges were indicated to best represent features that characterize the common diurnal behavior of $O_3$.

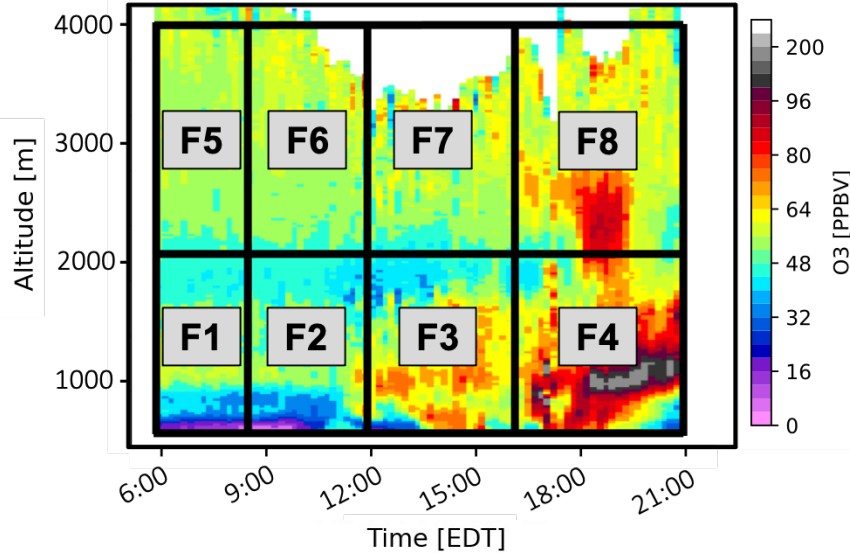


**Figure 2.** Clustering method developed for clustering vertical $O_3$ profiles taken from lidar measurements. The color coding

shows a typical day of lidar measurements of $O_3$ profiles on August 6, 2018, from the LMOL at Westport, CT during the
LISTOS Campaign. F1 – F8 indicate the time and altitude range of the eight features used for the clustering algorithm.

The features were evaluated for cluster tendency, essentially to confirm our dataset contained meaningful clusters

(discussed in detail in Sect. S2). One statistical approach was used to test the dataset called Hopkins statistic which measures
whether there is uniform distribution (spatial randomness) within the dataset (Lawson and Jurs, 1990). The results calculated
using the Hopkins statistic concluded a value higher than 0.75 (actual = 0.77) which by this standard indicates a clustering
tendency at the 90 % confidence level. Evaluating different feature options did not lead to better statistical results than with
the final chosen features. To visualize the cluster tendency of our dataset, we applied the algorithm of the visual assessment of
cluster tendency (VAT) approach (Bezdek and Hathaway, 2002) which uses the Euclidean distance measure to compute the
dissimilarity matrix in the dataset and creates an ordered dissimilarity matrix image. Figure S1 shows the VAT approach results
which indicates high similarity (red) and low similarity (blue) and confirms a cluster structure (not random) within our dataset.

Since the choice of clustering algorithm is subjective, we chose K-means clustering for its simplicity and widespread use.

To use the K-Means clustering algorithm, the optimal number of clusters based on your dataset must be chosen beforehand.
For this study, the package Nbclust (Charrad et al., 2014) in R was used, which applies 30 indices for determining the optimal
number of clusters. Using this package, as well as testing the quality of the clustering results using the silhouette method
(Kaufman & Rousseeuw, 1990), we selected six clusters as the optimal number of clusters. Since the K-Means clustering
algorithm is based on the Euclidean distance to each centroid, the input data was normalized (to a mean of zero and standard
deviation of one) to ensure each feature is given the same importance in the clustering (Aksoy & Haralick, 2001; Larose,
2005). The resulting six clusters (described fully in Sect. 3.1) represent clusters of regularly observed lidar $O_3$ curtains for the
regions of our study during the campaign periods.

**2.2.3 Missing data**
Although the input features were tailored based on the structure of the lidar measurements, the remaining data still had
missing data points. In performing a quick evaluation on the 8 input features (Figure S5), we found that Features 1, 4, 5, and
8 had the most missing data while Features 2, 3, 6, and 7 had few or zero cases of missing data. This means that the earlier
morning measurements (06:00 – 12:00 EDT) and the later evening measurements (16:00 – 21:00 EDT) had the most cases of
missing data points. This is plausible as the campaign teams were best able to retrieve clear measurement during
midday/evening hours (12:00 – 16:00 EDT). As a result, 51 out of 91 $O_3$ profile curtains had at least one missing data point
(feature) throughout the individual profile curtain.
A common practice for dealing with missing data is complete case analysis (CCA), in which observations with missing
values are completely ignored, leaving only the complete data to cluster. CCA can be inefficient as it introduces selection bias
since the sample data no longer retains the state of the original full dataset (Donders et al., 2006; Little & Rubin, 2014). When
we applied CCA, there were only 40 $O_3$ profile curtains of complete data, removing over half of the study profiles. Instead,
we used a more comprehensive solution – imputation - that yields unbiased results (Donders et al., 2006). For this study we
used the single imputation (SI) technique *knnImputation* in R (Torgo, 2010), which uses the k-nearest neighbors and searches
for the most similar cases and uses the weighted average of the values of those neighbors to fill the missing data. Essentially,
this method selects the days that have the most similar profile curtain to any profile which has missing data points and uses
those real data points to calculate a weighted mean that will fill in the missing data. We acknowledge using an imputation
method on the dataset will possibly introduce a bias which is difficult to quantify, but this allows the use of the full 91 profile
curtains of $O_3$ data. The silhouette method was used to test the quality of the newly imputed dataset and proved to be no worse,
nor better, than the CCA (*real data*) results. Therefore, the dataset was first imputed using SI to create a complete dataset and
then the clustering method described in the sect. before (2.2.2) was applied to the complete imputed dataset.

**2.3. Model simulations**
The offline GEOS-Chem chemical-transport model (CTM) was utilized to simulate the spatial and temporal variability
of coastal $O_3$ in the Chesapeake Bay and Long Island Sound during the time of the campaigns. The GEOS-Chem model is a
global 3-D CTM driven by assimilated meteorological data from the NASA Global Modeling and Assimilation Office
(GMAO). Our simulations were driven by reanalysis data from Modern-Era Retrospective analysis for Research and
Applications, Version 2 (MERRA-2; Gelaro et al., 2017). We ran a nested GEOS-Chem (v12-09) simulation at 0.5° x 0.625°
horizontal resolution over the eastern portion of North America and adjacent ocean (90 – 60°W, 20 – 50°N), using lateral
boundary conditions updated every three hours from a global simulation with 2° x 2.5° horizontal resolution. The nested
GEOS-Chem simulation was run with 72 vertical levels from 1013 to 0.01 hPa. Since the study focuses on the altitude range
0 – 4000 m, the first 20 vertical levels from GEOS-Chem were used with 14 levels within the boundary layer (≤ 2000 m). The
nested simulation was conducted for the study periods June – September 2017 and April – August 2018. We used the standard
"out-of-the-box" unmodified default settings from the tropospheric chemistry chemical mechanism (tropchem) with global
anthropogenic emissions from the Community Emissions Data System (CEDS) inventory (McDuffie et al, 2020) and U.S.
Environmental Protection Agency (EPA) National Emissions Inventory (NEI) 2011 for monthly mean North American
regional emissions (EPA NEI, 2015).
We also used results from NASA's near real-time forecasting system, GEOS-CF, an online GEOS-Chem simulation (v12-
0-1) from GMAO (https://gmao.gsfc.nasa.gov/-weather_prediction/GEOS-CF/) with GEOS coupled to the GEOS-Chem
tropospheric-stratospheric unified chemistry extension (UCX) and run at a high spatial resolution of 0.25°, roughly 25 km
(Keller et al., 2021, Knowland et al., 2021). The vertical resolution for GEOS-CF is interpolated onto 72 vertical levels from
1000 to 10 hPa. Since the study focuses on the altitude range 0 – 4000 m, the first 21 vertical levels from GEOS-CF were used
with 14 levels within the boundary layer (≤ 2000 m). Prior to the launch of the 12z five-day forecast, GEOS-CF produces daily
global, 3-D atmospheric composition distributions using the GEOS meteorological replay technique (Orbe et al., 2017), and
this study makes use of these historical estimates, made available to the public for the period since January 2018.  Therefore,
the GEOS-CF results shown in this study only include the dates from OWLETS-2 and LISTOS campaigns, since they both
occurred in 2018.
While both model simulations use similar versions of GEOS-Chem chemistry, there are noteworthy differences to keep
in mind during the analysis of the clustering. The main differences between the two models are (1) GEOS-Chem is an offline
CTM using archived meteorology, while GEOS-CF simulates atmospheric composition simultaneously with meteorology
(online); (2) the spatial resolution of the GEOS-CF model (0.25°) is higher than GEOS-Chem (0.5° x 0.625°); and (3) the
GEOS-CF model runs with Harmonized Gridded Air Pollution (HTAP; v2.2; base year 2010) anthropogenic emissions from
the Emission Database for Global Atmospheric Research (EDGAR), while GEOS-Chem was run with CEDS anthropogenic
emissions (base year 2014). These imperative differences can lead to disparities in the following results.

**3. Results & Discussion**
**3.1 Overview of the 2-D $O_3$ curtain clusters**
The clustering results reveal distinctive characterized $O_3$ behavior during the three campaigns in which $O_3$ concentrations
vary across the clusters. As previously mentioned in Sect. 2.2.3, the clustering analysis initially identified six cluster groups
from the $O_3$ profile curtains. Only one date was assigned to Cluster 6 (16 June 2018): the lidar profile curtain on this day
(Figure S6) shows a large fraction of data missing, and the available data have relatively high $O_3$ throughout the lowest 3 km,
which is different from other clusters. Therefore, we consider Cluster 6 to be an outlier and will not include it in the subsequent
analysis.
Various $O_3$ and surface meteorological parameter cluster statistics for the remaining five clusters are summarized in Table
1. With only 5 of the 2-D profile curtains assigned, Cluster 5 depicts the least common $O_3$ behavior during the campaigns. On

the other hand, Cluster 3 is the most common O$_3$ behavior during the campaigns with 28 profile curtains assigned to this cluster. Following Cluster 3, Cluster 1 is the next most common cluster with 25 profile curtains. Cluster 2 and Cluster 4 fall in the middle with 14 and 18 profile curtains assigned to the cluster numbers, respectively.

| Cluster # | a) No. of vertical profiles | b) O$_3$ Max (ppb) | c) O$_3$ Min (ppb) | d) T avg. (min; max) (°F) | e) WS avg. (min; max) (m s$^{-1}$) |
|---|---|---|---|---|---|
| 1 | 25 | 86.5 | 42.2 | 74.1 (67.8; 86.4) | 1.5 (0.5; 2.8) |
| 2 | 14 | 72.8 | 28.9 | 71.6 (64.0; 83.9) | 1.6 (0.6; 2.9) |
| 3 | 28 | 86.6 | 34.2 | 77.2 (67.0; 87.6) | 1.3 (0.5; 2.4) |
| 4 | 18 | 97.8 | 44.1 | 78.4 (68.0; 90.4) | 1.2 (0.4; 2.3) |
| 5 | 5 | 67.7 | 29.1 | 74.5 (66.8; 74.5) | 1.2 (0.3; 3.4) |

**Table 1.** Lidar vertical O$_3$ profile cluster statistics: a) total number of vertical profiles; b) O$_3$ maximum; c) O$_3$ minimum O$_3$;) AQS monitoring station cluster mean d) surface temperature and e) wind speed; minimum and maximums in parenthesis. The statistics and averages were derived from the total number of profile curtains assigned to each cluster.

The five clusters were distinguished by the varying O$_3$ concentrations between the low-level and mid-level as well as diurnal variations (Figure 3). Figure 3a quantifies the between-cluster differences. We separate the data by the two altitude subsets (low and mid-level) and by two time subsets (morning = 6:00 – 12:00 and afternoon = 12:00 – 21:00) for lucidity as the majority of the cluster differences are contrasted between these subsets. In the low-level, all five clusters exhibit the common O$_3$ diurnal pattern where surface O$_3$ is titrated overnight and reaches a minimum but then is quickly exacerbated with the increase of sunlight throughout the day and typically peaks after midday (Figure 3b). The extent of this common diurnal pattern varies by cluster.

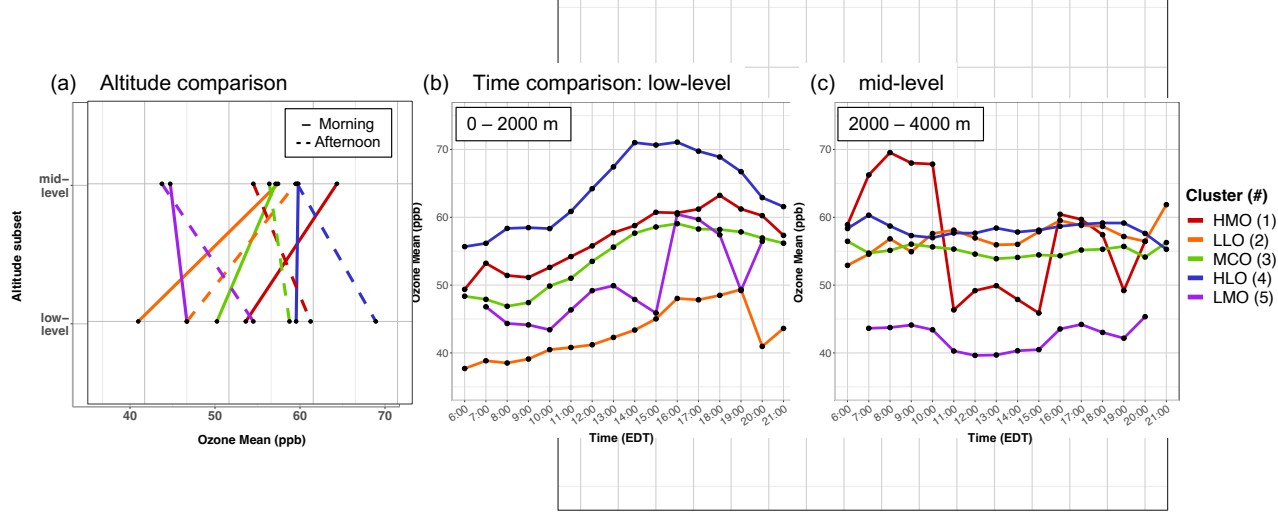

**Figure 3.** Lidar $O_3$ cluster average comparisons (five clusters depicted in colors). a) Altitude comparison of mean $O_3$ averaged over time: morning hours from 6:00 – 12:00 (solid line) and afternoon hours from 12:00 – 21:00 (dashed lines). Time comparison of mean hourly $O_3$ split between the b) low-level and c) mid-level.

Cluster 1 in the low-level has the second highest morning and afternoon $O_3$ average (52 and 59 ppb) and in the mid-level the highest morning $O_3$ average (64 ppb) (Figure 3a). Cluster 1 also exhibits the most unique pattern of mid-level $O_3$ (Figure 3c), with the highest concentrations found in the early morning and an uncharacteristic plunge to lower $O_3$ concentrations from 11:00 – 15:00 EDT. This is contrary to the other clusters which do not show much $O_3$ variation temporally in the mid-level. The majority of the individual profile curtains assigned to Cluster 1 show concentrated early morning residual layers in the mid-level that diffuse after the morning, which is distinctive to the other clusters. In the low-level, Cluster 2 has the lowest morning and afternoon $O_3$ average among the clusters (39 and 45 ppb) with moderate mid-level $O_3$ concentrations. Cluster 3 has the most uniform vertical $O_3$ extent between the low and mid-level (Figure 3a), in contrast to the other clusters that differ greatly in $O_3$ concentrations between the two altitude subsets. Cluster 4 has the highest morning and afternoon $O_3$ averages (59 and 68 ppb) in the low-level, reaching > 70 ppb temporally (Figure 3b). Finally, Cluster 5 has, considerably, the lowest morning and afternoon $O_3$ averages (42 and 43 ppb) in the mid-level, almost 10 ppb lower than the other clusters. Cluster 5 does not have a smooth-evolving $O_3$ diurnal pattern in the lower level (Figure 3b), which can be attributed to the averaging of only five different profile curtains that were assigned to this cluster (Table 1).

Figure 4a illustrates the mean lidar $O_3$ 2-D profile curtains for each of the clusters. For Cluster 1, 3, 4, and 5, higher $O_3$ concentrations in the low-level are captured during afternoon/evening time (12:00 – 21:00 EDT), with the highest low-level $O_3$ in Cluster 4 (> 70 ppb). This behavior follows the common diurnal pattern of $O_3$, that was distinguishable in Figure 3b. This common $O_3$ growth reaches vertically to approximately 1500 m for each of the clusters but is generally contained below 2000 m. Differing from the low-level $O_3$ behavior, mid-level $O_3$ is generally less variable in magnitude throughout the entire profile curtain (except for Cluster 1; see Figure 3a). The highest $O_3$ concentrations for the mid-level are exhibited in Cluster 1, 2, 3, and 4, with the highest mid-level $O_3$ in Cluster 1 during the early morning hours ($\geq$ 70 ppb).

Following the descriptions above, each cluster is given a nomenclature according to their unique characteristics. Cluster 1 is termed as the highest mid-level $O_3$ (HMO) cluster; Cluster 2 as the lowest low-level $O_3$ (LLO) cluster; Cluster 3 is the most common $O_3$ (MCO) cluster; Cluster 4 is the highest low-level $O_3$ (HLO); Cluster 5 is the least common and lowest mid-level $O_3$ (LMO) cluster. The $O_3$ variability represented and justified above is what led to the successful clustering of the lidar $O_3$ 2-D profile curtains.

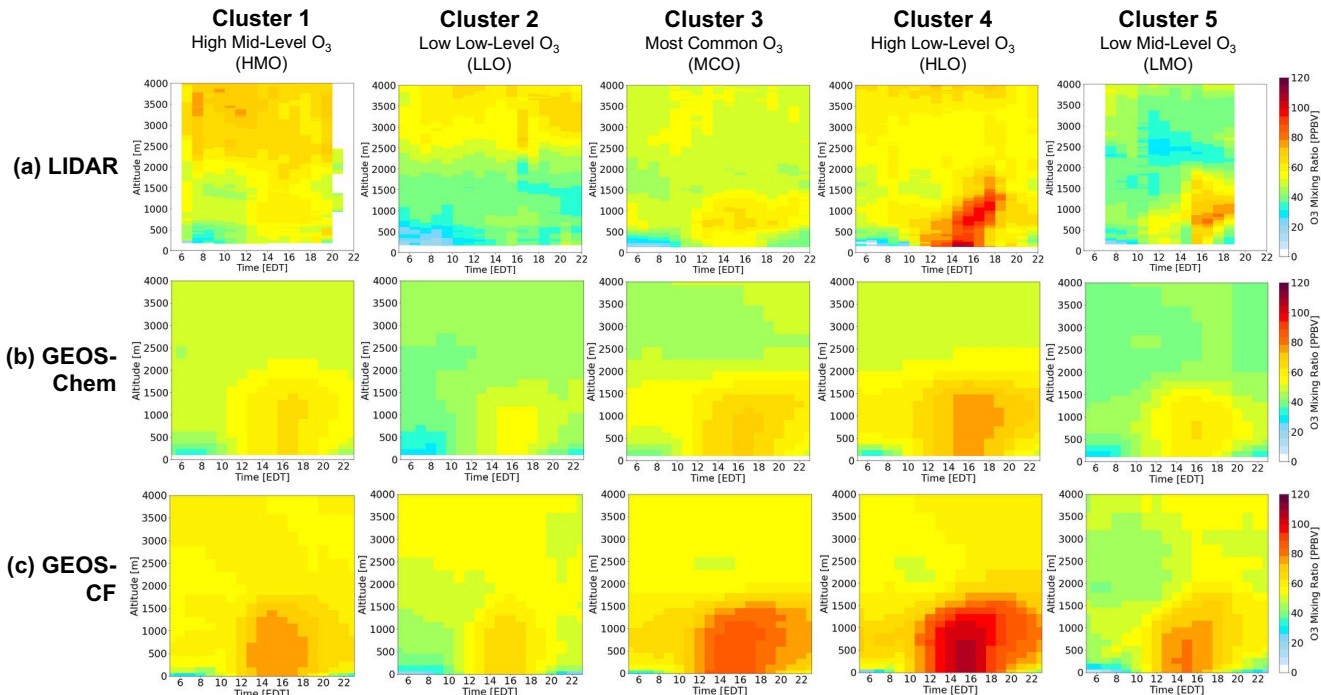

**Figure 4.** Cluster-mean $O_3$ vertical profile results by cluster assignment (1- 5) and arranged: a) LIDAR; b) GEOS-Chem simulation; and c) GEOS-CF simulation.

The clustering analysis results provided a characterization of $O_3$ behavior that transpired during these three campaigns. Figure 3b and 3c indicate each cluster represents a different $O_3$ evolution pattern, likely related to different photochemical or transport regimes. This kind of evaluation is useful in that it combines $O_3$ information from both temporal and vertical dimensions. For example, the HLO cluster reveals the specific case in which higher $O_3$ is captured early in the temporal profile in the low-level and translates to the higher $O_3$ captured in the low-level as well. The profile curtains show higher background $O_3$, indicating these cases did not have "clean air" to begin with which can allow a greater accumulation in the low-level in the afternoon. This is an example of how this type of clustering analysis, if applied, could demonstrate background $O_3$ in the similar case studies. In another example, several profile curtains assigned to the HMO cluster indicate concentrated residual layers in the mid-level and possible entrainment to the surface as the day progressed. To prove this feature, vertical velocity and vertical velocity variance data would be needed but the knowledge that a clustering approach is able to pinpoint these features that could only be discernible through lidar measurements proves to be useful. The clustering results was valuable in recognizing a significant large pollution related cluster (HLO), a total of 18 out of the 91 curtain profiles which correspond with the highest daily surface maxima measured at these sites (= 97.77 ppb) (Table 1). This cluster, on average, exhibited a daily surface maxima up to 10 ppb greater than any of the other clusters. Discerning these higher $O_3$ cases is imperative for mitigating severe air pollution.


## 3.2. Cluster surface analysis


To support the lidar clustering results, daily averaged meteorological surface observations from AQS stations pertaining
to the campaign period and GEOS-Chem surface model output were evaluated in regard to the five clusters. Figure 5 shows
the cluster mean surface temperature from AQS stations and GEOS-Chem model as well as the simulated wind speed and
direction. The average surface temperature from each station is represented as the circular markers while the simulated
temperatures are represented as the spatial contour and the simulated wind speed (m s$^{-1}$) and direction as arrows. Cluster
average, minimum, and maximum AQS surface temperature and wind speed can be found in Table 1d, e.

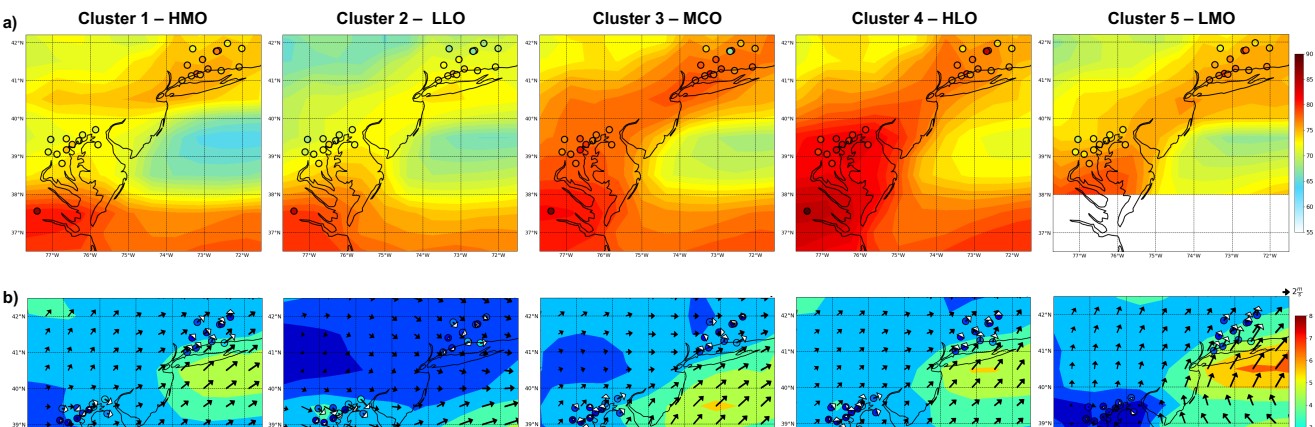


**Figure 5.** Cluster averaged meteorological surface AQS station observations and GEOS-Chem model results. a) Surface
temperature observations represented as the circular markers and simulated surface temperatures represented as the spatial
contour (top-panel). b) Surface wind speed and direction observations represented as the circular markers and white arrows
and simulated wind speed and direction represented as spatial contour and black arrows (bottom-panel).

In general, the surface meteorological conditions agree with our knowledge of transport and O$_3$ production that would
lead to each of the five clustered lidar O$_3$ profile curtains. It is evident that the clusters with the highest surface O$_3$ (HMO,
MCO, and HLO) all share a predominant offshore, westerly wind. Furthermore, MCO and HLO presented higher overall
observed and simulated surface temperatures compared to the other clusters (Figure 5a). Observed and simulated wind speeds
reveal slightly lower average wind speeds and primarily continental wind flow for both clusters as well (Figure 5b). These
meteorological conditions are conducive to a higher production of surface O$_3$ concentrations which validates the higher O$_3$
found in the low-level results (Figure 3b, 4a).
Conversely, the lowest surface temperatures are found in LLO. Lower surface temperatures are also indicative of low
vertical mixing due to less generation of convection. Relatively calm wind speeds and lower temperatures indicate other
possible meteorological factors such as high cloud cover that could have contributed to the lower $O_3$ concentrations in LLO.
Although surface $O_3$ concentrations in LMO reach higher levels later in the day, first at 13:00 EDT and then again at 16:00
EDT, the rest of the temporal profile stays below moderate levels. Average temperatures for LMO are moderately high but, in
contrast, the average wind speed is higher (specifically over the Long Island Sound) and unique to the other clusters, wind
direction is predominantly onshore (Easterly – Southerly). This prevalent onshore flow indicates a transport of cleaner marine
air which corroborates the lower surface $O_3$ levels. LMO did not have any profile curtains assigned from OWLETS-1 which
is why data for the lower Chesapeake Bay area is not shown in Figure 5.
There was only one occurrence during the dates in which the lidar instruments were operating in which there was a
recorded maximum daily 8-hour average (MDA8) $O_3$ exceedance (> 70 ppbv). This exceedance date is 25 May 2018 in which
3 AQS sites in the LISTOS region measured MDA8 $O_3$ of 73, 72, and 72 ppbv. This curtain profile was assigned to the HMO
cluster (Cluster 1), the cluster with high $O_3$ in the mid-level and moderate $O_3$ in the low-level and near the surface.

**3.3. Evaluating the GEOS-Chem and GEOS-CF model**
In this sect. the model results from GEOS-Chem and GEOS-CF will be compared to the lidar data using the five lidar $O_3$
profile clusters discussed in Sect. 3.1. Both model results were sampled in an equal manner, in which we extracted the same
cluster date assignments from the lidar clusters and created mean vertical profiles based on the model results. This allowed us
to evaluate the model performance based on the five characterized $O_3$ lidar clusters. As mentioned previously, the GEOS-CF
simulation data is not available for 2017. Thus, the results shown subsequently will only include GEOS-CF results from 2018
(only dates from the OWLETS-2 and LISTOS campaigns). The GEOS-Chem simulation results include both years thus all
three campaign duration periods.

**3.3.1 Overall model performance**
Figure 4b and 4c depict the simulated cluster-mean $O_3$ profile curtains from GEOS-Chem and GEOS-CF, mirroring the
mean lidar profile curtains in Figure 4a. For all clusters in the low-level, both models simulate a consistent accumulation of
$O_3$ near the surface after 12:00 EDT, mirroring the $O_3$ common diurnal pattern depicted in mean lidar profile curtains in Figure
4a. However, the extent the models simulate is often higher in magnitude than the observations, specifically GEOS-CF
consistently predicting the accumulation at a higher magnitude than GEOS-Chem. In the mid-level, both models simulate
much less $O_3$ variability than what is captured in the lidar observations. Figure 4b and 4c clearly show how the models struggle
to reproduce any mid-level $O_3$ pattern or variability that is relayed in the lidar observations. This is in contrast to the low-level
where the models are able to reproduce the common diurnal pattern of $O_3$. With the lidar data providing a full temporal and
vertical profile curtain of $O_3$ behavior and development, we are able to indicate areas where the models struggle such as in this
case in the mid-level.

We first evaluate overall correlation and biases between the model and lidar data. The overall correlation between both models and the lidar data, disregarding the specific clusters, based on the two altitude subsets as the performances differ between low-level and mid-level for both GEOS-Chem (Figure S7a) and GEOS-CF (Figure S7b). The mean normalized biases for the five clusters displayed in Table S1 (in Supplementary Material) were calculated from the total vertical and diurnal averages separated by low-level and mid-level. For both models, overall low-level $O_3$ correlation rounds to 0.70, signifying a strong relationship between the model simulations and the lidar observations (Figure S7 - top panel). This indicates that both models can simulate the development and pattern of $O_3$ well in the low-level. Overall, GEOS-Chem performs well in simulating low-level $O_3$ with a lower non-systematic normalized bias ranging from -0.10 to +0.13 for the five clusters. Thus, based on the lower bias, GEOS-Chem fairs well simulating the magnitude of low-level $O_3$ as well. For all clusters, GEOS-CF overestimates the average magnitude of low-level $O_3$ with a systematic high positive normalized bias ranging from +0.30 to +0.67. This consistently high bias reveals that GEOS-CF generally is unable to simulate low-level $O_3$ magnitude well.

For the mid-level, the overall correlation reveals that GEOS-CF and GEOS-Chem both have a weak relationship with the lidar (R = 0.22 and R = 0.12, respectively) (Figure S7 - bottom panel). This indicates that neither model is able to simulate mid-level $O_3$ pattern well. GEOS-Chem consistently underestimates the magnitude of mid-level $O_3$ with a systematic high negative normalized bias ranging from -0.44 to -0.18, for all clusters, while GEOS-CF has a lower and non-systematic normalized bias ranging from -0.22 to 0.28. Overall, both models are not able to simulate the variability of $O_3$ nor the magnitude well in the mid-level. The overall analysis in this sect. provides a fundamental but condensed assessment of model performance. In the next sect., the cluster specific differences reveal additional model performance insight that would be conceivably overlooked when evaluating overall performance.

### 3.3.2 Model evaluation based on lidar clusters

**Figure 6**. Mean profile curtain spatial $O_3$ difference (model – lidar observations) for each cluster (1 – 5). GEOS-Chem differences (a) and GEOS-CF differences (b).

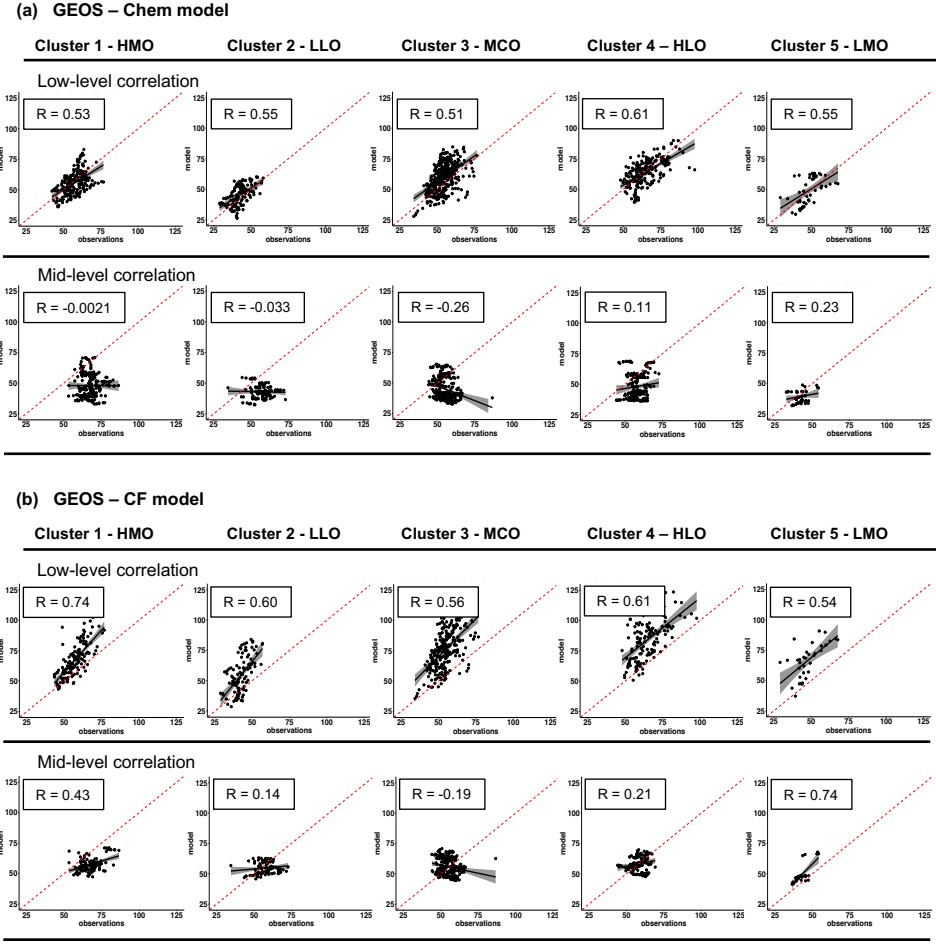

**Figure 7.** $O_3$ correlation between lidar observations and a) GEOS-Chem model simulation results and b) GEOS-CF model results by each cluster split by low-level (top panel) and mid-level (bottom panel).

Significant cluster by cluster differences are unmasked in evaluating the models based on the established $O_3$ behavior cases. To quantify the results illustrated in Figure 4, we show spatial $O_3$ differences (model – lidar observations) for each cluster (Figure 6) as well as individual cluster correlation (Figure 7) (subsequent cluster calculated normalized biases and correlation can be found in Table S1). Evaluating the individual cluster biases and correlation reveal more in-depth model discrepancies as well as areas where the models perform well.

In the low-level, GEOS-CF has a similar performance ability for the HMO, HLO, and LMO clusters with high positive biases at + 0.30, + 0.41, and + 0.45 respectively. These higher biases imply GEOS-CF has difficulty capturing moderate $O_3$ concentrations below 2000 m (HMO and LMO) as well as the in the high $O_3$ cases (HLO). GEOS-CF also has a high positive bias (+ 0.50) in the LLO cluster indicating that GEOS-CF struggles to capture the lower $O_3$ concentrations in the low-level.

This is warranted as models are intended to approximate and are not usually able to capture extremes (high or low) but GEOS-
CF also seems to struggle capturing moderate cases as well. In the low-level, GEOS-Chem has the best performance (minimal
-0.04 bias and strong correlation, R = 0.61) in HLO, which is the cluster with the highest low-level $O_3$ accumulation (refer to
Figure 4a). The second-best performance for GEOS-Chem in the low-level follows closely behind (minimal +0.07 bias and
fair correlation, R = 0.55) in LLO, the cluster with the lowest $O_3$ accumulation. These results suggest GEOS-Chem actually
performs well in cases of high $O_3$ as well as cases of low $O_3$ with a slight tendency to overpredict lower $O_3$ concentrations and
underpredict higher $O_3$ concentrations. This challenges the overall assumption that models struggle to capture extreme cases
since GEOS-Chem actually performs best in simulating both extreme cases of high $O_3$ in HLO and, again, low $O_3$ in LLO.
GEOS-Chem has a similar performance for the LMO and HMO clusters with negative biases of $-0.10$ and $-0.09$, respectively.
GEOS-Chem is also able to capture the moderate $O_3$ in both of these clusters well with slight underestimations.
Both models perform the worst (in comparison with the other clusters) in the low-level in the MCO cluster with a $+0.13$
bias for GEOS-Chem and $+0.67$ bias for GEOS-CF. As described in Sect. 3.1, MCO is the most common cluster with moderate
- high average $O_3$ concentrations in the low-level (refer to Figure 3b). Although GEOS-Chem has the worst performance in
the MCO cluster, it is not necessarily a poor performance. The performance follows the conclusion previously made that
GEOS-Chem can fairly simulate moderate $O_3$ in the low-level although, in this case, with slight overestimations. Contrarily,
the GEOS-CF performance in the MCO cluster reveals a more substantially high positive bias. This stands out as models are
usually able to capture moderate levels (e.g., non-extreme cases). Evaluating the full temporal and vertical profile indicates
that the higher GEOS-CF bias in the MCO cluster is additionally influenced by the greater overestimation of morning $O_3$, not
solely the afternoon $O_3$. This is different to the performance in the LLO and LMO clusters where GEOS-CF also had a high
positive bias in the low-level but does better simulating the early morning $O_3$ magnitude. A similar conclusion can be drawn
when evaluating the low-level GEOS-Chem performance. HMO, LLO, MCO, and LMO all share 'higher' biases (rounding to
$+/- 0.10$), but the highest bias is found in the MCO cluster. Analogous to GEOS-CF, this can similarly be attributed to GEOS-
Chem overestimating morning $O_3$ the worst in the MCO cluster in contrast to the better early morning estimation in the other
clusters.
In the mid-level, GEOS-Chem underestimates $O_3$ magnitude to the greatest extent in the HMO and the LLO cluster (both
bias = $-0.44$), which are both clusters with higher mid-level $O_3$ concentrations (refer to Figure 3c). GEOS-Chem performs
similarly in the HLO and MCO clusters, with a negative mean bias of $-0.30$ and $-0.27$, respectively. This indicates that
GEOS-Chem most struggles to simulate higher concentrations of $O_3$ in the mid-level. The GEOS-Chem model actually never
reaches $O_3$ cluster averages greater than 50 ppb, directly divulging the greater systemic negative bias in the mid-level. GEOS-
Chem simulates LMO mid-level $O_3$ magnitude the best ($-0.18$ bias), which is the cluster with the lowest $O_3$ average (< 45
ppb). Although for the LMO cluster GEOS-Chem has a lower bias, the correlation is still poor (R = 0.23) which indicates that
the model is relatively capable of simulating mid-level $O_3$ only when the case devises lower concentrations but still fails to
replicate any $O_3$ variability and pattern.

On the other hand, GEOS-CF does best simulating LLO, MCO, and HLO, which are all clusters with moderate $O_3$ in the mid-level ($\geq 50$ and $\leq 70$ ppb). GEOS-CF has the highest bias in the LMO cluster (+ 0.28), the cluster with the lowest mid-level $O_3$ magnitude. GEOS-CF also has the strongest correlation in the same LMO cluster (R = 0.74). This is a unique case where although GEOS-CF is not able to capture the magnitude in the mid-level, it is able to capture the pattern of low $O_3$ well. Comparing the full multi-dimensional lidar and model mean profile curtains it is evident that in the LMO cluster, the GEOS-CF model simulates a similar mid-level $O_3$ pattern in the early morning/afternoon that is captured in the mean lidar curtain profile. The second worst performance for GEOS-CF is the underestimation of mid-level $O_3$ in the HMO cluster, contrarily the cluster with the highest mid-level $O_3$ ($\geq 70$ ppb). This supports the previous conclusion that although GEOS-CF has a relatively lower biases in the mid-level, the model still struggles to simulate the extreme $O_3$ cases. Although GEOS-CF underestimates $O_3$ magnitude in the HMO cluster, it actually has a higher correlation than most of the other clusters (R = 0.43) (Figure 7, Table S1). In comparing the full multi-dimensional lidar and model mean profile curtain (Figure 3), GEOS-CF does a fair job connecting the mid-level higher $O_3$ pattern in the early morning that develops down to the low-level later in the afternoon. From this we can draw a conclusion that GEOS-CF is better able to capture mid-level $O_3$ patterns earlier in the temporal profile leading to higher correlations with the lidar.

### 3.3.3 Cluster approach and model conclusions

Several studies rely on case study investigations or grouping data by altitude to evaluate model performance. As demonstrated in Sect. 3.3.1, we can evaluate the overall summarized the model profile curtains $O_3$ against the lidar profile curtains and come to the simple conclusion that both models fairly simulate low-level $O_3$ but struggle to simulate mid-level $O_3$. However, a systematic and comprehensive understanding of the different photochemical regimes in coastal regions does not only require case studies and overall summaries. The clustering approach allows for a comprehensive yet still detailed evaluation of the different photochemical regimes in coastal regions utilizing the lidar derived full profile curtains. Additionally, using the clusters, we can efficiently evaluate the ability of the models to simulate many different cases of $O_3$. This approach revealed specific $O_3$ cases in which the models perform well and others where the models fail that would have been overlooked by solely considering the overall results. Using the clustering, we are able evaluate how the cluster specific differences (Figure 6, Figure 7, and Table S1) reveal additional model performance insight and specific gaps that would be conceivably overlooked when evaluating overall performance.

It is warranted that models struggle simulating extreme events/cases such as seen in the low-level in the HLO cluster and in the LLO cluster. However, GEOS-Chem performs best in both clusters with minimal biases and strong to fair correlations. Our result suggest that GEOS-Chem does a much better job simulating extreme $O_3$ cases in the low-level than expected. This specific model feature is not eminent when evaluating overall performance. Additionally, overall GEOS-Chem performs poorly in the mid-level. The detailed analysis granted by the cluster approach reveals GEOS-Chem has the lowest bias in the LMO cluster signifying the model is better able to capture low $O_3$ conditions in the mid-level. The overall high systemic positive bias for GEOS-CF in the low-level is further dissected when evaluating the individual clusters. GEOS-CF

systematically overestimates low-level $O_3$, but the individual clusters indicate that the model has a better correlation with $O_3$
in HMO cases. An even more profound case is exposed in which GEOS-CF has a strong correlation with mid-level $O_3$ in the
LMO cases despite having a low correlation overall. This concludes that in cases where the GEOS-CF model struggles to
reproduce $O_3$ concentrations, the model can still capture the $O_3$ variability seen by the lidar measurements.
The clustering approach also reveals more discrepancies in the models such as in the MCO cluster. The advantage of
evaluating full temporal and vertical profile curtains indicates that overestimation of early morning $O_3$ throughout the low-
level leads to the poorer performances in MCO for both models. The overestimation of morning $O_3$ in GEOS-CF adds to the
systemic overestimation in afternoon $O_3$ contributing the greater bias and poorer correlation. The same case can be found in
the GEOS-Chem MCO cluster performance but to a lesser extent as GEOS-Chem has a much lower positive bias. Previous
studies have found that excessive vertical mixing leads to overestimation of $O_3$ near the surface as well as underestimation of
$O_3$ night-time depletion resulting in overestimation of $O_3$ the next day (Dacic et al., 2020; Keller et al., 2021; Travis &
Jacob, 2019). The titration that occurs at night after the initial afternoon build up requires successful simulation to prevent the
model beginning the following day with higher $O_3$ than is observed which can lead to the overprediction of $O_3$ later that day.
Therefore, in the given case where there is an $O_3$ event that lasts more than one day (at the same lidar location), the model will
likely underestimate $O_3$ night-time depletion, overpredict morning $O_3$, and subsequently overpredict the afternoon build-up.
Given multiple cases of multi-day high $O_3$ events from the lidar measurements (17 total from HMO, MCO, and HLO), this is
likely one of the reasons for GEOS-CF overestimating early and therefore afternoon $O_3$ in these high $O_3$ cases in the low-level.
In Figure 6, GEOS-CF exhibits the greatest afternoon $O_3$ overprediction in MCO and HLO. In HLO alone, there were 4 (out
of 18) of the profiles that were consecutive while in MCO there were 8 (out of 28). This gives explanation for upwards of 22
– 29 % of the overestimation of $O_3$ in the profile curtains of these clusters. These multi-day $O_3$ events are particularly important
as they can indubitably lead the models to higher overprediction of afternoon $O_3$. As the full lidar profile curtains reveal, the
models tend to overestimate early morning $O_3$ in the MCO cases which links to the overestimation in afternoon $O_3$ as well.
Both models have a better ability to simulate early morning $O_3$ magnitude and pattern for other clusters than the MCO.
For example, GEOS-CF does best simulating morning low-level $O_3$ in cases of lower $O_3$ extent (LLO and LMO). Excluding
MCO, GEOS-Chem does not have such an issue overestimating low-level $O_3$ in the afternoon. In the other clusters, GEOS-
Chem actually underpredicts early morning low-level $O_3$ in the full vertical profile. An underestimation of early morning $O_3$
does not warrant the same build-up up of afternoon $O_3$. This gives some explanation to why GEOS-Chem underpredicts the
other clusters with higher $O_3$ concentrations in the low-level (HMO and HLO). In the mid-level GEOS-Chem has a systemic
high negative bias for all clusters, consistently underestimating $O_3$ but the clusters reveal a better performance in LMO, the
cluster with lowest mid-level $O_3$ extent. It is evident that the model cannot simulate cases with higher $O_3$ concentrations in the
mid-level but simulates low $O_3$ cases better. On the other hand, GEOS-CF results indicate a lower non-systemic bias in the
mid-level. Since the version of GEOS-Chem used in this study was run with the tropchem chemistry mechanism which
excludes stratospheric chemistry (now obsolete with current GEOS-Chem developments) and GEOS-CF uses the UCX
chemistry mechanism that includes stratospheric chemistry, this may allude to better performance of GEOS-CF in simulating

higher $O_3$ concentrations in the mid-level. Both models indicate weak correlations with the lidar observations in the mid-level and it is apparent that both models struggle to capture the pattern of $O_3$ behavior in the mid-level. This could be due to multiple model inefficiencies such as the coarse model resolutions. Although GEOS-CF has a finer resolution than GEOS-Chem, it still may not be sufficient in horizontal and vertical grid resolution to replicate the $O_3$ variations captured in the 2-D lidar observations.

There are additional model discrepancies that can lead to underestimations of $O_3$ in GEOS-Chem in the mid-level that was found in all 5 clusters. One gap in the GEOS-Chem model could be the representation of tropospheric halogen chemistry which has a large effect of coastal $O_3$ production. Newer updates to the GEOS-Chem model (v12.9) have included updated tropospheric halogen chemistry mechanisms (iodine, bromine, and chlorine) (Wang et al., 2021). This study found that the updated halogen chemistry actually worsens the overall underestimation of $O_3$ throughout the troposphere, specifically in the northern hemisphere, indicating further investigation of halogen chemistry is needed for better model representation. Another study finds a similar conclusion in the proper representation of cloud uptake and tropospheric chemistry (Holmes et al., 2019). This study found that implementing an updated, more accurate, and stable cloud entrainment-limited uptake in the GEOS-Chem model reduces the sensitivity of oxidants and aerosol chemistry in the troposphere but still had little effect on $O_3$ model comparison to observations (such as sonde and aircraft). This is due to the environmental variability being much higher than the effect of $NO_x$ and $O_3$ cloud chemistry but still warrants further testing. The role lightning plays in tropospheric oxidation is another feature that is commonly misrepresented in global models and can affect $O_3$ simulation (Mao et al., 2021). These are all examples of features that if not simulated correctly can lead to misestimations of $O_3$. The clustering approach allows us to organize the detailed lidar measurements to scope out specific cases where these misrepresentations occur. These previous studies also highlight the importance of lidar measurements and their ability to depict tropospheric emission development and behavior throughout the vertical profile and diurnal cycle which can be used to constrain model emissions and improve simulations.

Although this analysis proves to be a useful technique to characterize the largely variably $O_3$ behavior in coastal regions and evaluate the subsequent model performance, there are also limitations. In this study we are comparing single point lidar versus model output, therefore we cannot simply state that the model is incorrect. We make conclusions and draw biases based on the ability to subset a grid point and compare that to a single point lidar curtain to the best ability but that still leaves an uncertainty. The high vertical and spatiotemporal resolution reveal intricate details about the behavior of $O_3$ during these campaigns. $O_3$ lidars have a unique advantage, compared to traditional surface measurements, in measuring vertical distribution of $O_3$ with respect to time. This advantage is of great value when investigating model ability in simulating the spatial and temporal distribution of $O_3$ and can provide crucial information in understanding surface $O_3$ events.

**3.4 Cluster derived case studies to evaluate modeled wind and ozone**

Meteorological factors such as wind speed and direction can directly impact whether a coastal region will experience clean air or $O_3$ exceedances. When local meteorological processes such as sea/bay breeze occur at such a fine scale, equally

fine resolution measurements are essential in capturing this. The Doppler wind lidar offers a focus on fine details that are only
revealed in the multi-dimensional data which allows for such a comprehensive evaluation of the established $O_3$ cluster profile
curtains. In this sect., we evaluate the 2-D relationship between wind and $O_3$ to assess model performance using lidar and
model derived profile curtains (Figure 8). We derived two specific case studies, each from a different cluster: MCO = 17 June
2018 and HLO = 30 June 2018. Utilizing the derived clusters, the case studies were chosen to focus on high low-level $O_3$
behavior cases with a goal of evaluating possible sea/bay breeze events. The two case studies are both from the HMI location
during the OWLETS-2 campaign. There are consistent Doppler lidar measurements throughout the low-level (< 2000 m) which
allows for a direct comparison with the simulated profiles; therefore, the focus of the following analysis will be on the low-
level altitudes. The deficit of mid-level observed wind data disallows for a conclusive and concrete evaluation of simulated
mid-level $O_3$.

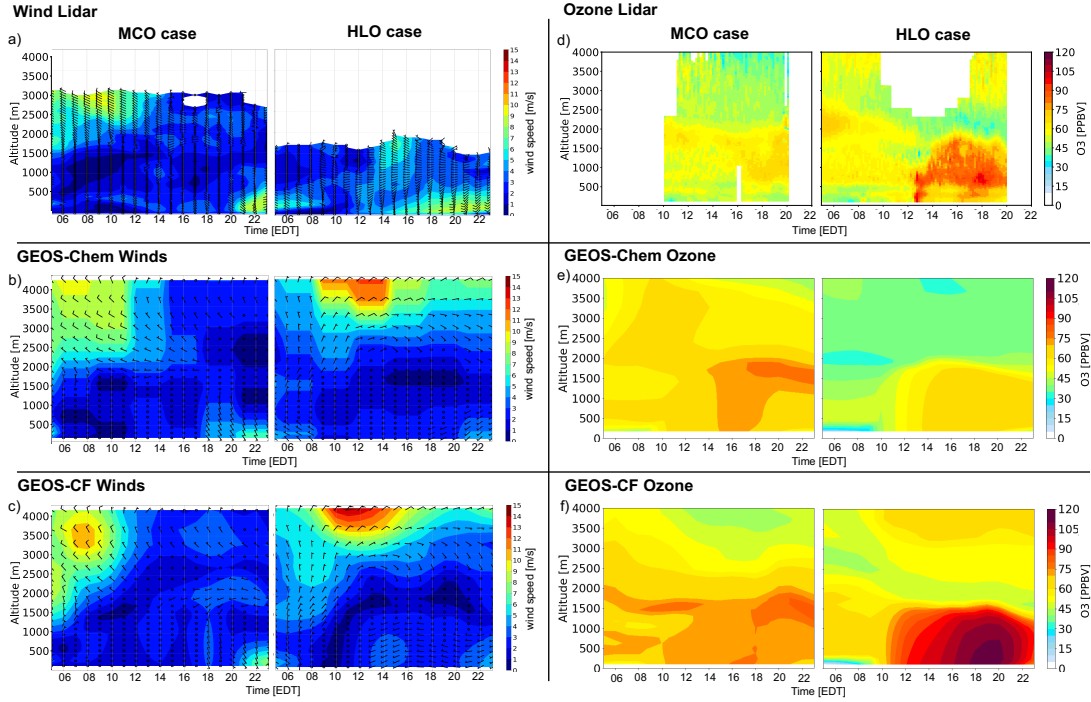


**Figure 8.** Profile curtains of wind speed/direction (a-c) and ozone (d-f) from the lidar (top panel), GEOS-Chem (middle panel),
and GEOS-CF (bottom panel). Results from OWLETS-2 at HMI.

**3.4.1 Sea breeze event interpretation**
GEOS-Chem and GEOS-CF both struggle to capture low-level wind speed and direction in both MCO and HLO cases
(Figure 8a-c). In the MCO case, the Doppler wind lidar captures a wind direction shift from westerly to easterly winds
beginning at 06:00 EDT accompanied by calm winds (approximately 0 m s$^{-1}$) indicating a likely common sea/bay breeze event.
The timing of the start of this event is simulated well but the models fail to predict an actual well-defined wind shift, instead
merely simulating 0 m s$^{-1}$ winds after 05:00 EDT. It is apparent that the models struggle to capture the finer processes such as
a sea/bay breeze which could have likely led the underprediction of wind speed. It is important to note that GEOS-Chem runs
with offline meteorology, averaged every 3 hours. Since sea/bay breezes often happen at a finer temporal resolution, the GEOS-
Chem model is at a disadvantage in modelling such fine processes. A wind direction shift is also depicted in the HLO case,
with westerly winds early in the morning and a shift to south-easterly winds later in the temporal profile (at about 10:00 EDT).
This could also likely be an early onset sea breeze event which could have contributed to the high observed O$_3$ concentrations
in the afternoon. Again, the exact timing of the start of the wind shift is captured by the models but then no defined directional
shift and little to no winds are simulated after. Both the MCO case and HLO case observe increased wind speeds near the
surface, first before 08:00 EDT then again in the evening. Both models underestimate the extent of the increased wind speeds.

**3.4.2 Relation to ozone cases and clustering**
In this sect., the wind lidar curtains will be assessed in relation to the O$_3$ lidar profile curtains and the model performance.
The results in sect. 3 revealed that both models had the highest bias and lowest correlation simulating low-level O$_3$ in MCO.
Evaluating the wind and O$_3$ lidar profile curtains against the model simulations helps paint a better picture as to why. Similar
to the MCO cluster mean curtain profile, early morning low-level O$_3$ in each case is overestimated by both models (Figure 8e,
f). There is higher O$_3$ captured in the lidar curtain profile, but it is constrained between 1500 – 2000 m. Both models bring this
higher O$_3$ pattern down to the surface (below 500 m) overestimating O$_3$ throughout the low-level. Since both models predict
little to no winds during this time, this could contribute to overestimations of O$_3$ near the surface.
In the HLO case, GEOS-CF overestimates low-level O$_3$ while GEOS-Chem underestimates low-level O$_3$. From sect. 3.3
the results revealed that although GEOS-CF has a high positive normalized bias for low-level O$_3$ in HLO, the model had a
reasonable relationship (R = 0.61) with the O$_3$ lidar measurements. This is corroborated with the individual HLO case (Figure
8f) as GEOS-CF is better able to simulate the development of O$_3$ in the low-level, especially in the early morning. The GEOS-
CF modeled winds mirror this performance with a better reproduction of the wind shift in HLO (Figure 8c). While GEOS-
Chem has a lower normalized bias for low-level O$_3$ in the HLO cluster, GEOS-Chem consistently underestimates wind speed
and fails to reproduce any wind shifts. This reveals that in the possible sea breeze event, the two models do not perform equally.
Since GEOS-Chem is an offline CTM using archived meteorology and GEOS-CF simulates atmospheric composition
simultaneously with meteorology (online), the replication of a sea breeze case would not necessarily be comparable.
In most cases, sea/bay breeze events can contribute to high concentrated daytime O$_3$ events in which O$_3$ is recirculated
throughout the region. Such cases would likely lead to a similar curtain profile as seen in the HLO case (Figure 8a), where
high O$_3$ in the morning is likely associated with the higher O$_3$ at the surface in the afternoon. But it is apparent that the cases
for MCO and HLO are dissimilar. We would expect per the clustering approach that sea breeze cases would most likely be
assigned to the same cluster, but this is not the case here. Investigating the full lidar and model profile curtains for the two
cases gives us more information as to why these two curtains are not in the same cluster. It is evident that the HLO case has
much higher afternoon O$_3$ near the surface (below 1000 m) than the MCO case, with peaks > 75 ppb at both 12:00 and again
at 16:00 EDT. In contrast, the MCO case has higher afternoon $O_3$ concentrations captured above 2000 m than the HLO case.
The HLO case has high $O_3$ in the afternoon, but it is constrained to the lower 2000 m and just above this high $O_3$ plume, there
is an $O_3$ deficit of almost 50 ppb. Although the MCO case also reveals lower $O_3$ above 2000 m, the vertical gradient in this
case is not as stark. This is also replicated in both models which simulate lower $O_3$ directly above the high surface $O_3$ in the
HLO cluster but simulate much higher $O_3$ above 2000 m in the MCO cluster. From their distinct vertical and temporal behavior,
it is easy to conclude why these two cases were not assigned to the same cluster.

The cases elected for MCO and HLO give reason to address the difficulty simulating complex coastal mechanisms.

Despite the fact that MCO and HLO both indicated prospective sea/bay breeze cases, the results of the simulated winds and
$O_3$ were distinctive. Simulating complex sea/bay and land relations is imperative for correctly mitigating high $O_3$ cases. To
accurately simulate such complex exchanges, high resolution vertical and horizontal simulations are needed. Because of the
models' relatively coarse resolutions (nominally 50 and 25 km horizonal resolution; 72 vertical levels), the fine-scale vertical
wind gradients and horizontal wind shifts are difficult to resolve and, in these cases, not fully able to replicate. This study also
acknowledges the need for an evaluation of other modeled factors, such as divulged in sect. 3.3.3, considering the possible
confounding effects on modeled $O_3$ outcome.

**4. Conclusion**

We developed and tested a clustering method on a suite of 91 multi-dimensional lidar $O_3$ profile curtains retrieved

from three recent land/sea campaigns (OWLETS-1, OWLETS-2, and LISTOS), during the summer months of 2017 and 2018.
The K-Means clustering algorithm, driven by 8 well defined features, was applied to categorize the fine resolution $O_3$ data,
revealing five distinct $O_3$ behavior cases that are distinct in pattern and magnitude vertically and temporally. We present five
different clusters of $O_3$ behavior identified as: highest mid-level $O_3$ (HMO) cluster; lowest low-level $O_3$ (LLO) cluster; most
common $O_3$ (MCO) cluster; highest low-level $O_3$ (HLO); lowest mid-level $O_3$ (LMO) cluster. The results indicate that fine
resolution data can be used to differentiate the behavior of $O_3$ in a region and classify different cases of $O_3$ exploiting the
multiple dimensions. The clustering approach allowed us to characterize the range of highly variable vertical and temporal
coastal $O_3$ behavior for the duration of these campaigns which can be a good indicator of how $O_3$ behaves in general in these
coastal regions during the summer months. Furthermore, this approach could be used by states to better identify different $O_3$
photochemical regimes and frequency beyond just surface sampling.

We evaluated the performance of two CTMs, GEOS-Chem and GEOS-CF, in these complex environments. Overall,

the models have the greatest difficulty simulating the vertical extent and variability of $O_3$ concentrations in the mid-level, with
weak overall relationships to the lidar observations (R = 0.12 and 0.22). GEOS-Chem had a systematic high negative bias and
GEOS-CF an overall lower unsystematic bias range. In the low-level, GEOS-Chem had overall low unsystematic bias range
and fair relationship with the lidar observations (R = 0.66), while GEOS-CF had a systematic high positive bias but overall
fair relationship (R = 0.69).
Utilizing the curated clusters reveals new model insight that is neglected in the overall performance analysis. The
cluster approach divulges specific model limitations but also cases in which the models perform well. GEOS-Chem simulates
low-level $O_3$ cases best in the HLO and LLO clusters and the worst in the MCO cluster. HLO and LLO are the clusters with
the most extreme (low and high) $O_3$ cases while MCO is the most common cluster with moderate $O_3$. This concludes that
GEOS-Chem does best simulating extreme low-level $O_3$ but struggles to capture the frequently occurring moderate $O_3$
behavior. GEOS-CF also has the greatest overestimations for low-level $O_3$ in the MCO cluster. Evaluating the full profile
curtain reveals that this overestimation can be most attributed to the greater overestimation of early morning $O_3$. This feature
is unique to the MCO cluster and warrants further investigation as $O_3$ left in the residual layer can contribute to higher $O_3$ in
the afternoon and proves to be a challenge for CTMs. The value of lidar measurements is reflected in its ability to reveal these
features.
Both models share poor performances in the mid-level but there are specific cases that stand out in the clustering
results, specifically the LMO cluster, in which GEOS-CF shares a good agreement with the lidar measurements. It can be
concluded that although the model struggles to simulate $O_3$ magnitude, it can relatively emulate the mid-level $O_3$ pattern in
LMO. This is also apparent in the MCO cluster, in which the pattern of higher mid-level $O_3$ that suggests a relationship with
the low-level $O_3$ is simulated fairly in the GEOS-CF model. This pattern is also a rare feature that is captured in the lidar that
demonstrates the significance of the measurements. The greater underestimations of mid-level $O_3$ for GEOS-Chem can be
alluded to multiple model discrepancies. Since the GEOS-Chem version and mechanism used in this study (tropchem) only
considers tropospheric chemistry we can expect the performance in the mid-level to have deficiencies. Although GEOS-CF is
run with the combined tropospheric and stratospheric chemistry mechanism, has a better grid resolution, and is an online
model, there are still limitations to both models especially when simulating mid-level $O_3$. Known model errors and coarse
horizontal and vertical grid resolution contribute to the difficulty in simulating fine-scale coastal $O_3$ variability. There are many
contributing model factors that can be affecting the performance of GEOS-Chem and GEOS-CF that were mentioned in this
study not solely coarse model resolution.
A unique value of the clustering approach on multi-dimensional lidar data is that it offers a convenient way to ascertain
different $O_3$ case studies. An example of this is our evaluation of two cases studies from the MCO and HLO clusters. Modeled
winds were evaluated using Doppler wind lidar data observed during the OWLETS-2 campaign. The wind lidar data was
mostly limited to lower altitudes (< 2000 m), which allowed for wind speed and direction validation at the low-level. The
morning wind deceleration and directional shifts (onshore to offshore) illustrated in lidar profile curtains indicate a possible
sea/bay breeze event in both case studies. This is likely another contributor that led to enhanced surface $O_3$ in these cases. Due
to the coarser model resolution, GEOS-Chem and GEOS-CF were not able to capture the sea breeze phenomena in these cases
which could have facilitated in the high $O_3$ biases for these clusters. With GEOS-CF having a finer horizontal resolution than
GEOS-Chem, the results reveal minimal advantages simulating the pattern of wind speeds better but none in simulating the
wind directional shifts. This affirms that the spatial resolution of GEOS-CF (~25 km) is still not fine enough for mesoscale
processes such as the sea/bay breeze. Although a regional model analysis is out of the scope of this study, we propose to use

multi-dimensional lidar measurements to evaluate finer regional modeling in our future work. We acknowledge that other factors, aside from model resolution, contribute to discrepancies in modeled coastal $O_3$ and further warrant a deeper evaluation. The clustering approach on lidar measurements offers an unmatched ability to pinpoint these features.

This work is the first time that all three associated campaign lidar data have been analyzed in conjunction. In utilizing the highly detailed suite of multi-dimensional lidar data, we are able to comprehensively explore the behavior and variability of coastal $O_3$ for the duration of the campaigns. Applying the clustering analysis directly to the lidar $O_3$ data emerges as a useful and robust approach for identifying $O_3$ patterns during the highly polluted summer months in coastal environments. Since the time of the OWLETS and LISTOS campaigns, the lidar instrument systems have been updated and are now more fully automatized for use eliminating such constraints faced in this study. Further observations using lidar instruments should be especially valuable in investigating coastal $O_3$ behavior as it can divulge the finer-scale $O_3$ characteristics that remain difficult to successfully simulate in CTMs. The time-height and fine resolution measurements only available from multi-dimensional lidar instruments were vital in allowing us to form these conclusions.

This kind of evaluation allows for detailed model assessment of specific $O_3$ cases that are unmasked through the clustering analysis. Looking at the overall correlations, it would seem the models have a good relationship with the low-level lidar observations but looking into the cluster-by-cluster differences, the gaps within the models are elucidated. Using the cluster assignments, we are able evaluate how the cluster specific differences reveal additional model performance insight that could be conceivably overlooked when evaluating overall performance. This work is a middle ground between looking at specific cases (or dates) and summarizing overall model performance. Additionally, the clustering approach provides an abridged way to detecting distinctive case studies. We provide a new approach that allows a synopsis of summer coastal $O_3$ behavior and subsequently model performance without completely muting distinct $O_3$ features. Evaluating model performance for diverse $O_3$ behavior in coastal regions is crucial for improving the simulation and furthermore, mitigation of air quality events.

*Code availability.* Model code is available upon request to the first author.

*Data availability.* The GEOS-Chem model simulation data from this study is publicly accessible online at https://doi.org/10.7910/DVN/V99LHT. The GEOS-CF model data is publicly available online at their website https://gmao.gsfc.nasa.gov/-weather_prediction/GEOS-CF/. The lidar data is publicly available online at https://www-air.larc.nasa.gov/missions.htm.

*Supplement.*

*Author contributions.* CB and YW conceived the research idea. CB wrote the initial draft of the paper and performed the analyses and model development. All authors contributed to the interpretation of the results and the preparation of the paper.

*Competing interests.* The authors declare that they have no conflict of interest.
*Acknowledgements.* This study is supported by NASA MUREP Graduate Fellowship (80NSSC19K1680). The Ozone Water-
Land Environmental Transition Study (OWLETS-1, 2) and Long Island Sound Tropospheric Ozone Study (LISTOS) field
measurements described here were funded by the NASA's Tropospheric Composition Program and Science Innovation Fund
(SIF), Maryland Department of Environment, the National Oceanic and Atmospheric Administration (NOAA), the
Environmental Protection Agency (EPA), the Northeast States for Coordinated Air Use Management (NESCAUM), and the
New Jersey and Connecticut Departments of Energy and Environmental Protection. The authors acknowledge the principal
investigators and data operators John Sullivan, Joel Dreessen, Ruben Delgado, William Carrion, and Joseph Sparrow as well
as the guidance of the Tropospheric Ozone Lidar Network (TOLNet). LMOL and TROPOZ data are publicly available at
(https://www-air.larc.nasa.gov/missions/TOLNet/). The OWLETS and LISTOS data are available at (https://www-
air.larc.nasa.gov/). The Doppler wind data taken from the UMBC wind lidar and are publicly available at (https://www-
air.larc.nasa.gov/cgi-bin/ArcView/owlets.2018). The GEOS-CF model simulation data were provided directly from the NASA
Center Global Modeling and Assimilation Office (GMAO) at the Goddard Space Flight Center
(https://gmao.gsfc.nasa.gov/weather_prediction/GEOS-CF/).

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
