# Peer review of "Cluster-based characterization of multi-dimensional tropospheric"

_Atmospheric Chemistry and Physics, 2022_

## Referee Comment (RC1)

Review of Bernier et al., 2022

This paper provides a unique method for taking advantage of ozone lidar data for evaluating models. This could be a valuable contribution and appropriate for ACP, however the description of this method and the model evaluation need major revision. As presented, the model evaluation using lidar data does not suggest anything to improve about the models other than increased resolution. Overall, the paper is lacking in explanations for the model failures in representing the five ozone clusters. If model resolution is the only clear factor, the authors should apply their method to regional, high-resolution modeling done for LISTOS (see comments below) at least. The model evaluation also does not provide much insight into how the clustering method is superior to a simple average comparison of model vs. observations. This study would be significantly strengthened by clear examples of how their clustering example provides specific insights over a simple average comparison. The authors could also better describe the benefits of lidar data beyond other types of profiles (e.g. ozone sondes, aircraft profiles) and show specific examples of these benefits.

**Major comments.**

*Clustering algorithm*

The k-means clustering algorithm needs better explanation. Did the authors choose the 8 features and then confirm they best represented the data with k-means? Did they try different numbers of features? What was the rational for looking at the data in this way? What about day to day variability, driven by different synoptic? I do not understand how the clustering algorithm was applied to the eight features. The authors might consider a diagram that shows how the 8 features lead to the 5 clusters.

*Analysis*

In the discussion of modeled vs. observed meterology, key findings should be discussed, not 'slight differences'. The readers are interested in what your model/observational comparison reveals about missing model processes that would be important to simulating surface ozone, particularly exceedances. Focus on highlighting those results and remove discussion about minor features.

*Model selection*

As the authors conclusion is that neither model has sufficient resolution to capture the seabreeze, this study would greatly benefit from including the regional modeling done for LISTOS (and possibly OWLETS?). WRF-Chem modeling was done for LISTOS. I suggest contacting NESCAUM to ask for this output.

**Minor comments.**

Line 58 – You say "set out to address this issue" twice.

Line 165 – Could you please further explain this sentence "Input features (seed values) were rationally established.."

Line 223 – Can you describe whether using the 40 complete profiles before datafilling was performed would give similar results? It seems somewhat problematic to fill the data based on observed patterns and then cluster the results also on observed patterns. Giving a little more information on why this approach is valid would be useful.

Line 288 – By temporal variation, do you mean diurnal variation?

Line 299 – Discuss Fig. 3a first or switch the order of the panels in Fig. 3.

Fig. 3a – It would be more informative to separate the profiles with altitude into day and night, or 12pm vs. 6am. Fig. 4 shows how the observations and models are both better mixed between roughly 12-16 EDT than during other hours.

Line 310 – Can you examine each of the 5 curtains and tell us for sure whether this is the reason? Or you could provide a standard deviation version of Figure 4 that would help us understand the cluster variability.

Line 319 – Give the cluster definitions earlier in the text before the introduction of Table 1.

Line 329 – Are the clusters spread across the three campaigns? Describe how each campaign contributes to each cluster.

Table 1 – Consider including Tmin and Tmax, and WSmin and WSmax. They could just be in parenthesis instead of separate columns.

Line 330 – What do you mean by "…could demonstrate background O3 in the case studies"?

Line 373 – When you show comparisons to the lidar data for GEOS-CF, are you only including lidar data clusters from OWLETS 2 & LISTOS?

Section 3.3.1 - The authors should use surface ozone monitors to determine whether ozone exceedances of the NAAQS occurred during any of the clusters. This would provide greater weight to the analysis of poor model performance for a given cluster.

Line 381 – The statement "In Figure 6, we first evaluate the overall relationship and correlation between both models and the lidar data, disregarding the specific clusters" is confusing as Fig. 6 is split into the 5 specific clusters.

Figure 6 – By "Spatial O3 difference", are you just referring to the differences in the vertical and in the diurnal cycle?

Line 403 – Do the models simulate higher ozone due to insufficient vertical resolution and/or excess vertical mixing? Is there anything to be learned in the only large model underestimate at the surface (GEOS-Chem, HLO)?

Line 404 – Are the lidar observations averaged to the model vertical (and temporal) resolution?

Line 427 – Is this a typo, do you mean "positive percent biases at 13.9, 18.9, and 19.7 %" instead of "positive percent biases at 0.139, 0.189, and 0.197 %? The other bias % values also look like they might be decimal values that need a 100 multiplier.

Line 448 – You state "Using the clustering, we are able evaluate how the cluster specific differences reveal additional model performance insight that would be conceivably overlooked when evaluating overall performance." Please give actual examples of how clustering is better than just "to simply group data by altitude to achieve a summarized model evaluation." A clear description of the benefits of clustering over the approach would greatly improve this discussion.

Line 464 – Does the model overestimate ozone on the first day of these multi-day events?

Figure 7 – Please make the limits on the x and y axes the same and add a 1-1 line.

Line 487 – Models are not intended to simulate 'intricate details', but rather the patterns that lead to high/low ozone at the surface. Could you rephrase to discuss how lidar data can contribute to that effort? What can the lidar data provide that surface ozone and sonde data cannot, and give specific examples of why this matters.

Line 498 – What "additional model performance insight" have you given us? Be more specific.

Figure 8 – Why not show a difference plot similar to Fig. 6?

Line 621 – As these models were run with emissions not provided specifically for the years 2017 and 2018, it might be more informative to look at normalized bias patterns as opposed to absolute biases. As this study is attempting to use the lidar to uncover areas of poor model performance with a focus on coastal meteorology, this approach would remove the impact of emissions not suited to the simulation year.

Line 631 – Throughout the paper, 'slightly' better results are not worth describing. Please focus on the most important, high-level results. For example, the finding that the models perform most poorly against the most common cluster is useful. Why do the models do better in cases other than the MCO? Is it because the sea-breeze is the most common pattern and this is most difficult for the models to capture? If so, this is a useful finding and should be more clearly stated.

Line 659 – You state, "Using the cluster assignments, we are able evaluate how the cluster specific differences reveal additional model performance insight that could be conceivably overlooked when evaluating overall performance." Be specific about the insights you have revealed. The current manuscript is not clear about what the major findings are from the manuscript, nor what the most relevant conclusions are for air quality models.

Data availability – The authors need to provide the data links for the observational data used in this study. The authors could also consider providing their clusters as a data product for model evaluation.

---

## Referee Comment (RC2)

This manuscript presents the analysis of 91 Lidar measurements of ozone across three different field campaigns along the eastern US coast during the summers of 2017 and 2018 via a clustering analysis and model simulations. The goal of this manuscript was to investigate the characteristics of generalized coastal ozone behavior and to assess the ability of model simulations to reproduce ozone in complex coastal areas. The authors use a K-means clustering algorithm driven by 8 features to cluster their ozone measurements into 5 behavior cases, which they analyzed for different meteorological events that help to describe the ozone behavior in each case. To evaluate model ability to reproduce coastal ozone behavior, two chemical transport models (GEOS-Chem and GEOS-CF) were used to simulate these same events. Both models struggled to capture high ozone concentrations, especially in the mid-level altitudes, and GEOS-CF tended to overestimate low-altitude ozone. Much of this model inability to capture these ozone events was attributed by the authors to an inability to successfully capture changing wind speeds and directions.

This manuscript is a unique analysis and within the scope of ACP. I think it could be a good addition to the literature with some revisions, which I have detailed below.

**Major comments:**

- The clustering analysis methods needs to be laid out in more depth. Please give a brief description of what a K-means clustering algorithm is, along with most things described in the methods section – Hopkins statistic, silhouette method, etc. How are the best number of clusters chosen? Is this based on variance or something similar? There must be some mathematical model behind the decisions you have made, and this would be helpful to include in the supplemental, along with statistical information that led you to your choice of the number of clusters.

- For the underestimate of ozone in the free troposphere by GEOS-Chem, please include further discussion of model updates that may contribute to this beyond just the lack of the UCX mechanism. Here are papers I suggest reading for more information on the low bias in recent versions of the GEOS-Chem model: halogen chemistry (Wang et al., 2021, doi: https://doi.org/10.5194/acp-21-13973-2021); NOy reactive uptake in clouds (Holmes et al., 2019, doi: https://doi.org/10.1029/2019GL081990); lightning-produced oxidants (Mao et al., 2021, doi: https://doi.org/10.1029/2021GL095740).

- The difference in correlations between GEOS-CF and GEOS-Chem (0.69 vs. 0.66) is not large enough to be of any statistical note (both round to 0.7). I suggest removing any discussion of this small difference in correlations. Statistically, the models perform the same.

**Minor comments:**

Line 57: "set out to address this issue" appears twice in the sentence. Please delete one.

Line 66-69: Ozonesondes are also able to resolve vertical levels. How does the vertical resolution of the lidar measurements compare to ozonesondes? What advantages do lidars have over sondes?

Line 264: "significantly" Please provide a p-value or level of significance.

Line 406: "modeled versus lidar observation spatial O3 differences" This is confusingly worded. The "differences" implies that you are performing a mathematical operation (model – observed), but what you are doing is plotting model vs observations in Figure 7. Please reword to better express that.

Line 589: Is the underestimate in wind speed and failure to reproduce wind shifts by GEOS-Chem explained by the use of offline meteorology? Some variables in MERRA-2 are averaged every 3 hours, and I assume that sea/bay breezes occur more rapidly than that.

Line 652 and 145: "Automatized" should be automated.

**Figures:**

Figure 5. Can the observed winds be added to this plot, perhaps as a second row of panels?

Figure S1. Remove Table 2 title from the top of the table.

**References**

Holmes, C. D., Bertram, T. H., Confer, K. L., Graham, K. A., Ronan, A. C., Wirks, C. K., and Shah, V.: The Role of Clouds in the Tropospheric $NO_x$ Cycle: A New Modeling Approach for Cloud Chemistry and Its Global Implications, Geophys. Res. Lett., 46, 4980–4990, https://doi.org/10.1029/2019GL081990, 2019.

Mao, J., Zhao, T., Keller, C. A., Wang, X., McFarland, P. J., Jenkins, J. M., and Brune, W. H.: Global Impact of Lightning-Produced Oxidants, Geophys. Res. Lett., 48, https://doi.org/10.1029/2021GL095740, 2021.

Wang, X., Jacob, D. J., Downs, W., Zhai, S., Zhu, L., Shah, V., Holmes, C. D., Sherwen, T., Alexander, B., Evans, M. J., Eastham, S. D., Neuman, J. A., Veres, P. R., Koenig, T. K., Volkamer, R., Huey, L. G., Bannan, T. J., Percival, C. J., Lee, B. H., and Thornton, J. A.: Global tropospheric halogen (Cl, Br, I) chemistry and its impact on oxidants, Atmospheric Chem. Phys., 21, 13973–13996, https://doi.org/10.5194/acp-21-13973-2021, 2021.

---

## Author Response (AR1)

**Reply to Reviewers**

We thank the two reviewers for their constructive comments and suggestions, which have further improved the quality of our manuscript considerably. Their comments are reproduced below with our responses in blue. The corresponding edits in the manuscript are highlighted with tracked changes. The line numbers are based on the revised manuscript.
* * *
**Reviewer #1:**

This paper provides a unique method for taking advantage of ozone lidar data for evaluating models. This could be a valuable contribution and appropriate for ACP, however the description of this method and the model evaluation need major revision. As presented, the model evaluation using lidar data does not suggest anything to improve about the models other than increased resolution. Overall, the paper is lacking in explanations for the model failures in representing the five ozone clusters. If model resolution is the only clear factor, the authors should apply their method to regional, high-resolution modeling done for LISTOS (see comments below) at least. The model evaluation also does not provide much insight into how the clustering method is superior to a simple average comparison of model vs. observations. This study would be significantly strengthened by clear examples of how their clustering example provides specific insights over a simple average comparison. The authors could also better describe the benefits of lidar data beyond other types of profiles (e.g. ozone sondes, aircraft profiles) and show specific examples of these benefits.

**Major comments.**

Clustering algorithm

The k-means clustering algorithm needs better explanation. Did the authors choose the 8 features and then confirm they best represented the data with k-means? Did they try different numbers of features? What was the rational for looking at the data in this way? What about day-to-day variability, driven by different synoptic? I do not understand how the clustering algorithm was applied to the eight features. The authors might consider a diagram that shows how the 8 features lead to the 5 clusters.

**Response:** We described the choosing and evaluation of the features for cluster tendency in lines 189 – 197. Following the reviewer's suggestion, we have further elaborated on the description of input features in the Supplementary Material Text S1. Clustering algorithm and efficacy tests are described in Text S2. Figure S2, S3, and S4 were added to help illustrate the clustering method.

One of the main factors that led the choosing of the features was the actual structure of the lidar measurements as described in Section 2.2.1. We had to work within the limitations of the lidar measurements e.g., the time of day they lidar instrument was operating and the processing of the data that can limit the data that is available as well as the accuracy of the data. We also had the goal to evaluate lower-level tropospheric ozone which is also why we limited the entire vertical profile to 4000 meters.

We chose the two altitude subsets based on the structure of the vertical atmosphere. One way we could have explored different input features would be to choose features that followed the

development of the boundary layer more precisely. Since a more in-depth evaluation of the development of the boundary layer would be needed to do this, we concluded that it was out of the scope of this work. Therefore, we chose the altitude subset 0 – 2000 meter to represent the complete evolution of the boundary layer, while the 2000 – 4000 meter subset represents the part of the vertical profile in which other factors such as longer range transport of pollutants would be of greater influence. We chose the 4 subsets of time following the common diurnal pattern of pollutant behavior. Tropospheric ozone has a common diurnal pattern that is greatly influenced by the presence of sunlight. The first subset of time (F1, F5) represents the early morning before the sunlight has reacted with precursor pollutants to create tropospheric ozone. The second time subset (F2, F6) represents the time of day in which the sun rising, and morning traffic have begun which both have an influence on ozone chemical reactions. The third subset of time (F3, F7) represents the midday time in which the sun is at its' full peak. At this time of the day tropospheric ozone usually peaks and remains at the maximum concentration of the day. The final time subset (F4, F8) represents the evening time in which the sun has/begun to set, and ozone concentrations decrease. As explained in line 167, we rationally chose these features to best represent the structure of the lidar measurements and to best represent the behavior of $O_3$ vertically and temporarily. With the goal of clustering most efficiently, this is in part to simply the data so that the results of the clusters are not weakened by too many input features. But also, not oversimplify the data so that we lose the details of the lidar data.

Analysis
In the discussion of modeled vs. observed meteorology, key findings should be discussed, not 'slight differences'. The readers are interested in what your model/observational comparison reveals about missing model processes that would be important to simulating surface ozone, particularly exceedances. Focus on highlighting those results and remove discussion about minor features.
**Response:** We agree with the reviewer's comment and have altered the manuscript to remove slight differences and to solely focus on larger discrepancies. See updated Section 3.3.1. We thank the reviewer for their constructive comment as the quality of the manuscript has been substantially improved.

Model selection
As the authors conclusion is that neither model has sufficient resolution to capture the sea breeze, this study would greatly benefit from including the regional modeling done for LISTOS (and possibly OWLETS?). WRF-Chem modeling was done for LISTOS. I suggest contacting NESCAUM to ask for this output.
**Response:** We have taken the reviewer's suggestion into consideration but have ultimately concluded that we will not be able to add a regional model analysis to the manuscript. The main purpose of this manuscript is NOT to evaluate specific model performances against LISTOS and OWLETS observations. Instead, our goal was to emphasize the value of the developed clustering method as an efficient way of comparing a suite of lidar observations with models. Also, the purpose of the manuscript is not to completely resolve specific model performance gaps but rather to highlight how the multi-dimensional lidar and cluster approach can help identify the model gaps more readily and provide more in-depth model insight. Per the reviewer's comments on providing a more comprehensive and detailed analysis of why this clustering approach can be more useful than a simpler method, we have better tailored the purpose of the updated

manuscript – see changes made in Section 3.3. Although a regional modeling analysis is out of the scope of the current manuscript, we added in the conclusion section that using multi-dimensional lidar measurements to evaluate regional modeling (such as WRF-Chem) will be in our future work (Lines 696 -697).

**Minor comments.**

Line 58 – You say "set out to address this issue" twice.
**Response:** Thank you, we corrected this. Line 57.

Line 165 – Could you please further explain this sentence "Input features (seed values) were rationally established…"
**Response:** We have further elaborated on the input feature selection in the Supplementary Material Text S1- Description of input features. As well as the clustering algorithm in Text S2.

Line 223 – Can you describe whether using the 40 complete profiles before data-filling was performed would give similar results? It seems somewhat problematic to fill the data based on observed patterns and then cluster the results also on observed patterns. Giving a little more information on why this approach is valid would be useful.
**Response:** We understand the reviewer's concern, and we have provided more clarification. This was mentioned in line 226: "The silhouette method was used to test the quality of the newly imputed dataset and proved to be no worse, nor better, than the CCA (*real data*) results."

We did test the cluster quality of the 40 complete profiles versus the imputed dataset. The 40 complete profiles did not reveal to have a better quality of clusters than the imputed dataset. Imputation is a common method that has been used for missing data before the clustering analysis. The single imputation technique provides weighted averages of values of the neighbors which is an accurate description of ozone spatially and temporally. An example: a case in which at 13:00 LT there is measured high ozone yet at the next time step there is absolutely no ozone is not common. The imputation was also applied to already averaged data. Therefore, the averaged data might be more representative of the case. We also acknowledged that using an imputation method will possibly introduce bias, but that bias is arduous to quantify. In testing the quality of the CCA dataset, we found that the significance of the clusters was no higher than that of the imputed dataset. Also, if we use the CCA dataset, we are eliminating over half of the curtain profiles with only 40 profiles of full data to use. We could argue that only using half the dataset to characterize coastal ozone during these campaigns introduces an equal or greater bias than imputing the individual curtain profiles to have a full dataset to work with. We therefore concluded to use the imputed dataset so that we could utilize all the full lidar profiles.

Line 288 – By temporal variation, do you mean diurnal variation?
**Response:** Yes. Line 295 changed to "diurnal variations" for clarification.

Line 299 – Discuss Fig. 3a first or switch the order of the panels in Fig. 3.
**Response:** The figure is initially discussed in totality to describe the differences of the clusters beginning at Line 294. We have updated the next line to more clearly mention Figure 3a first.

Line 295 – 297: Figure 3a quantifies the between-cluster differences. We separate the data by the two altitude subsets (low and mid-level) and by two time subsets (morning = 6:00 – 12:00 and afternoon = 12:00 – 21:00) for lucidity as the majority of the cluster differences are contrasted between these subsets.

The next paragraph describes the cluster differences more in depth but in order of cluster number.

Fig. 3a – It would be more informative to separate the profiles with altitude into day and night, or 12pm vs. 6am. Fig. 4 shows how the observations and models are both better mixed between roughly 12-16 EDT than during other hours.
**Response:** This is a good point as the profiles are very different in the morning hours versus afternoon. We have updated Figure 3a (be

[Figure]

**Figure 3.** Lidar $O_3$ cluster average compa                                                    tude comparison of mean $O_3$ averaged over time: morning hours from 6:00 – 12:00 (solid line) and afternoon hours from 12:00 – 21:00 (dashed lines). Time comparison of mean hourly $O_3$ split between the b) low-level and c) mid-level.

Line 310 – Can you examine each of the 5 curtains and tell us for sure whether this is the reason? Or you could provide a standard deviation version of Figure 4 that would help us understand the cluster variability.
**Response:** This line in the manuscript has been changed to better describe Cluster 5's low-level uniqueness.

Line 318-320: "Cluster 5 does not have a smooth-evolving $O_3$ diurnal pattern in the lower level (Figure 3b), which can be attributed to the averaging of only five different profile curtains that were assigned to this cluster (Table 1).

We cannot say that Cluster 5 merely has the most variable low-level $O_3$ since it actually has a slightly lower standard deviation than Cluster 4. This is because Cluster 4 has the highest evening $O_3$ and compared to the rest of the diurnal pattern such as the early morning $O_3$, this is the most variable. By making changes to lines 318-320 to better indicate the description of Cluster 5 and since standard deviation statistics do not bring any new information that cannot be derived from Figure 4 already, we have decided to leave the figure as is without adding the standard deviations.

Line 319 – Give the cluster definitions earlier in the text before the introduction of Table 1.
**Response:** We understand the reviewer's suggestion, but since the cluster nomenclature are derived based on the paragraphs before this (i.e., the entire descriptions of the different clusters), we do not think that the cluster definitions can be introduced before Table 1. Even so, the information that is provided in Table 1 (e.g., Table 1a – No. of vertical profiles) is also used in the naming of the clusters. To remove the confusion, we have eliminated the nomenclature of the clusters from Table 1 and have just left the cluster numbers.

Line 329 – Are the clusters spread across the three campaigns? Describe how each campaign contributes to each cluster.
**Response:** Yes, the different campaigns are spread throughout the clusters. The different campaigns do not contribute uniquely to any cluster.

Table 1 – Consider including Tmin and Tmax, and WSmin and WSmax. They could just be in parenthesis instead of separate columns.
**Response:** Done.
We have updated Table 1:

| Cluster # | a) No. of vertical profiles | b) $O_3$ Max (ppb) | c) $O_3$ Min (ppb) | d) T avg. (min; max) (°F) | e) WS avg. (min; max) (m s$^{-1}$) |
|---|---|---|---|---|---|
| 1 | 25 | 86.5 | 42.2 | 74.1 (67.8; 86.4) | 1.5 (0.5; 2.8) |
| 2 | 14 | 72.8 | 28.9 | 71.6 (64.0; 83.9) | 1.6 (0.6; 2.9) |
| 3 | 28 | 86.6 | 34.2 | 77.2 (67.0; 87.6) | 1.3 (0.5; 2.4) |
| 4 | 18 | 97.8 | 44.1 | 78.4 (68.0; 90.4) | 1.2 (0.4; 2.3) |
| 5 | 5 | 67.7 | 29.1 | 74.5 (66.8; 74.5) | 1.2 (0.3; 3.4) |

Line 330 – What do you mean by "...could demonstrate background O3 in the case studies"?
**Response:** The clustering analysis captures an associated trend of background $O_3$ (Cluster 4 – HLO), which reveals the profile begins with the highest averaged ozone. For HLO cluster the higher $O_3$ present in the mid-level also translates to the higher $O_3$ found in the low-level near the surface. This could mean that this higher $O_3$ already present in the early hours may not be a part of the local production of $O_3$. This could be helpful in other analyses looking for cases of background $O_3$ versus locally produced $O_3$ cases. I clarified this statement in the manuscript for further understanding. Updates below.

Line 339 - 345: Figure 3b and 3c indicate each cluster represents a different $O_3$ evolution pattern, likely related to different photochemical or transport regimes. This kind of evaluation is useful in that it combines $O_3$ information from both temporal and vertical dimensions. For example, the HLO cluster reveals the specific case in which higher $O_3$ is captured early in the temporal profile in the low-level and translates to the higher $O_3$ captured in the low-level as well. The profile curtains show higher background $O_3$, indicating these cases did not have "clean air" to begin with which can allow a greater accumulation in the low-level in the afternoon. This is an example of how this type of clustering analysis, if applied, could demonstrate background $O_3$ in the similar case studies.

Line 373 – When you show comparisons to the lidar data for GEOS-CF, are you only including lidar data clusters from OWLETS 2 & LISTOS?
**Response:** Yes, the quantified biases are only based on OWLETS-2 and LISTOS lidar data for GEOS-CF. But referring to Figure 4a, the mean lidar curtain profiles in comparison with the models, includes the average cluster profiles from all campaigns, so it includes OWLETS-1 as well.

Section 3.3.1 - The authors should use surface ozone monitors to determine whether ozone exceedances of the NAAQS occurred during any of the clusters. This would provide greater weight to the analysis of poor model performance for a given cluster.
**Response:** This is a great suggestion. The exceedance analysis is added to the manuscript.
Lines 385 – 388: There was only one occurrence during the dates in which the lidar instruments were operating in which there was a recorded maximum daily 8-hour average (MDA8) $O_3$ exceedance (> 70 ppbv). This exceedance date is 25 May 2018 in which 3 AQS sites in the LISTOS region measured MDA8 $O_3$ of 73, 72, and 72 ppbv. This curtain profile was assigned to the HMO cluster (Cluster 1), the cluster with high $O_3$ in the mid-level and moderate $O_3$ in the low-level and near the surface.

Line 381 – The statement "In Figure 6, we first evaluate the overall relationship and correlation between both models and the lidar data, disregarding the specific clusters" is confusing as Fig. 6 is split into the 5 specific clusters.
**Response:** This is a typo. Figures 6 and 7 were accidentally switched and have been fixed. Additionally, the correct Figure 6 has been moved to the Supplementary Material as Figure S7 and the original Figure S4 has been moved to the manuscript as Figure 7.

Line 410 - 412: We first evaluate overall correlation and biases between the model and lidar data. The overall correlation between both models and the lidar data, disregarding the specific clusters, based on the two altitude subsets as the performances differ between low-level and mid-level for both GEOS-Chem (Figure S7a) and GEOS-CF (Figure S7b).

Figure 6 – By "Spatial O3 difference", are you just referring to the differences in the vertical and in the diurnal cycle?
**Response:** Yes. The spatial $O_3$ difference between the full mean profile curtain of the model and the lidar observations.

Line 404 – Are the lidar observations averaged to the model vertical (and temporal) resolution?
**Response:** The lidar observations are interpolated to the resolution of the model in Figure 6.

Line 427 – Is this a typo, do you mean "positive percent biases at 13.9, 18.9, and 19.7 %" instead of "positive percent biases at 0.139, 0.189, and 0.197 %? The other bias % values also look like they might be decimal values that need a 100 multiplier.
**Response:** The reviewer is correct, this is a typo.

Per the reviewer's suggestion (for Line 621) subsequently to calculate the biases with normalized data not absolute data, we have update Table S1 again (see comment and changes further later in the reviewer's comments).

Line 448 – You state "Using the clustering, we are able evaluate how the cluster specific differences reveal additional model performance insight that would be conceivably overlooked when evaluating overall performance." Please give actual examples of how clustering is better than just "to simply group data by altitude to achieve a summarized model evaluation." A clear description of the benefits of clustering over the approach would greatly improve this discussion.
**Response:** We have updated the analysis in Section 3.3.2 to include a more in-depth analysis of the model performance insight that is only perceivable through the cluster-by-cluster differences.

Line 464 – Does the model overestimate ozone on the first day of these multi-day events?
**Response:** Correct, the model overestimates ozone on the first day as well but the model also overestimates ozone the rest of the days during the multi-day events.

Figure 7 – Please make the limits on the x and y axes the same and add a 1-1 line.
**Response:** Done.

As mentioned previously, Figure 6 and Figure 7 were accidentally switched in the submitted manuscript, but we corrected this.

New Figure 6 below:

[Figure]

Also updated Figure S4 with same additions:

[Figure]

(a) GEOS – Chem model

(b) GEOS – CF model

With the current changes to the manuscript per the reviewer's suggestion to elaborate on cluster specific model insight, we have shifted focused from overall evaluation. Therefore, the previous Figure 6 (above) was moved to the Supplementary Material as Figure S7 and the original Figure S4 has been moved to the manuscript as Figure 7. These changes were made to shift more focus on the different cluster specific model evaluation.

Line 487 – Models are not intended to simulate 'intricate details', but rather the patterns that lead to high/low ozone at the surface. Could you rephrase to discuss how lidar data can contribute to that effort? What can the lidar data provide that surface ozone and sonde data cannot and give specific examples of why this matters.

**Response:** The manuscript has been updated in Section 3.3.3 to further elaborate on the value of lidar data and its contribution to model simulation. The lidar data provides full temporal but possibly most importantly, vertical observations of $O_3$ data. This data gives us a fuller story of how $O_3$ behaves and develops throughout the day and throughout the altitude range. Since we are evaluating full multi-dimensional model curtains, we want to interpretate the vertical profile therefore not only focusing on the surface level performance. Another example that is mentioned in the manuscript explains how the clustering method proves to be more useful in finding cases where concentrated residual layer in the mid-level could have possible entrainment to the low-level/surface. This kind of feature would only be discernable through the use of lidar measurements as $O_3$ is not only developing from the mid-level to the low-level, but it is developing specifically over a period of time (please refer to Lines 345-346). Although this type of evaluation would need more data to support, the lidar data provides a full characterization of temporal and vertical distribution of $O_3$ that cannot be provided by other measurements. Another

specific example of the importance of using lidar data to evaluate the full temporal and vertical can be found in assessing the GEOS-CF performance in the LLO and LMO cluster. With the multi-dimensional curtain profile, we can indicate that GEOS-CF simulates early morning $O_3$ very well throughout the low-level. This is different than all the other clusters in which the model overestimates morning $O_3$ in the low-level. This point is discussed in more detail in the updated Section 3.3.2 and 3.3.3 in the manuscript.

Line 498 – What "additional model performance insight" have you given us? Be more specific.
**Response:** We fully agree with the reviewer that the discussion needs to be more specific. We have added a more in-depth evaluation of the cluster-to-cluster differences and the value that the clustering approach and lidar measurements bring in Section 3.3.2. and conclusions to Section 3.3.3. In these updated sections we have elaborated on specific model insight that the cluster approach has revealed that is not apparent in the overall model evaluation.

Figure 8 – Why not show a difference plot similar to Fig. 6?
**Response:** Although this would indeed be helpful in evaluating the wind speed differences, because of the intricacy of the wind arrows on the plots, it is not as feasible to do a difference plot of Figure 8. We would reason to keep the individual curtain wind profiles so that the wind direction differences are easily gaged.

Line 403 – Do the models simulate higher ozone due to insufficient vertical resolution and/or excess vertical mixing? Is there anything to be learned in the only large model underestimate at the surface (GEOS-Chem, HLO)?
**Response:** GEOS-Chem actually does a fair job estimating the high $O_3$ in the HLO cluster with only a -0.04 normalized bias. This shows us that GEOS-Chem is able to simulate the extreme cases of high low-level $O_3$. The model performance conclusions were updated (Section 3.3.2 and 3.3.3).

Line 621 – As these models were run with emissions not provided specifically for the years 2017 and 2018, it might be more informative to look at normalized bias patterns as opposed to absolute biases. As this study is attempting to use the lidar to uncover areas of poor model performance with a focus on coastal meteorology, this approach would remove the impact of emissions not suited to the simulation year.
**Response:** We agree with the reviewer's suggestion and have updated the analysis to evaluate normalized bias patterns as opposed to absolute biases. Table S1 was updated with the new calculated values and the manuscript was updated with the changes. See the new _final_ Table S1 below:

Table S1. Mean normalized bias & Correlation coefficient (R)

| a) GEOS-Chem Low-level | Cluster 1 - HMO | Cluster 2 - LLO | Cluster 3 - MCO | Cluster 4 - HLO | Cluster 5 - LMO |
|---|---|---|---|---|---|
| Bias | - 0.10 | 0.07 | 0.13 | - 0.04 | - 0.09 |
| R | 0.53 | 0.55 | 0.51 | 0.61 | 0.55 |
| b) GEOS-Chem Mid-level | | | | | |
| Bias | - 0.44 | - 0.44 | - 0.27 | - 0.30 | - 0.18 |
| R | - 0.002 | - 0.033 | - 0.26 | 0.11 | 0.23 |
| c) GEOS-CF Low-level | | | | | |
| Bias | 0.30 | 0.50 | 0.67 | 0.41 | 0.45 |
| R | 0.74 | 0.60 | 0.56 | 0.61 | 0.54 |
| d) GEOS-CF Mid-level | | | | | |
| Bias | - 0.22 | - 0.07 | 0.05 | 0.02 | 0.28 |
| R | 0.43 | 0.14 | - 0.19 | 0.21 | 0.74 |

Line 631 – Throughout the paper, 'slightly' better results are not worth describing. Please focus on the most important, high-level results. For example, the finding that the models perform most poorly against the most common cluster is useful. Why do the models do better in cases other than the MCO? Is it because the sea-breeze is the most common pattern and this is most difficult for the models to capture? If so, this is a useful finding and should be more clearly stated.
**Response:** We agree with the reviewer's comment and have removed cases of 'slightly' better results. We find that with the MCO cluster, the full profile curtain mean reveals a high estimation of early morning $O_3$. Higher estimated early $O_3$ can lead to higher afternoon estimations. Additionally, the MCO case in Section 3.4, does consider the possible sea breeze affect leading to higher afternoon estimations of $O_3$. The wind profile curtain reveals that winds are underestimated in this case which again, also leads to overestimations. These conclusions are divulged in Lines 459 – 472 and in Lines 634 – 637.

Line 659 – You state, "Using the cluster assignments, we are able evaluate how the cluster specific differences reveal additional model performance insight that could be conceivably overlooked when evaluating overall performance." Be specific about the insights you have revealed. The current manuscript is not clear about what the major findings are from the manuscript, nor what the most relevant conclusions are for air quality models.
**Response:** We have augmented the results and discussion in Section 3 revealing more cluster specific differences as well as supporting the value of the cluster and lidar approach. We also removed any small differences which skewed the scope of the manuscript. The focus is now channeled on the value of the method and what it reveals for air quality models that can be overlooked when evaluating overall performance. Please refer to the updated Section 3.3.2 and Section 3.3.3 for the subsequent changes.

Data availability – The authors need to provide the data links for the observational data used in this study. The authors could also consider providing their clusters as a data product for model evaluation.
**Response:** The GEOS-Chem model simulation from this study was made publicly available upon submission of the manuscript at https://doi.org/10.7910/DVN/V99LHT. The lidar data is publicly available at https://www-air.larc.nasa.gov/missions.htm (Line 100 - 101). The cluster data can be available upon request.
* * *
**Reviewer #2:**
This manuscript presents the analysis of 91 Lidar measurements of ozone across three different field campaigns along the eastern US coast during the summers of 2017 and 2018 via a clustering analysis and model simulations. The goal of this manuscript was to investigate the characteristics of generalized coastal ozone behavior and to assess the ability of model simulations to reproduce ozone in complex coastal areas. The authors use a K-means clustering algorithm driven by 8 features to cluster their ozone measurements into 5 behavior cases, which they analyzed for different meteorological events that help to describe the ozone behavior in each case. To evaluate model ability to reproduce coastal ozone behavior, two chemical transport models (GEOS-Chem and GEOS-CF) were used to simulate these same events. Both models struggled to capture high ozone concentrations, especially in the mid-level altitudes, and GEOS-CF tended to overestimate low-altitude ozone. Much of this model inability to capture these ozone events was attributed by the authors to an inability to successfully capture changing wind speeds and directions.

This manuscript is a unique analysis and within the scope of ACP. I think it could be a good addition to the literature with some revisions, which I have detailed below.

**Major comments:**

- The clustering analysis methods needs to be laid out in more depth. Please give a brief description of what a K-means clustering algorithm is, along with most things described in the methods section – Hopkins statistic, silhouette method, etc. How are the best number of clusters chosen? Is this based on variance or something similar? There must be some mathematical model behind the decisions you have made, and this would be helpful to include in the supplemental, along with statistical information that led you to your choice of the number of clusters.

  **Response:** Following the reviewer's suggestion, we have further elaborated on the clustering algorithm in the Supplementary Material Text S2 - Description of clustering algorithm and cluster efficacy tests. Figure S2, S3, and S4 were added to illustrate the clustering method.

- For the underestimate of ozone in the free troposphere by GEOS-Chem, please include further discussion of model updates that may contribute to this beyond just the lack of the UCX mechanism. Here are papers I suggest reading for more information on the low bias

in recent versions of the GEOS-Chem model: halogen chemistry (Wang et al., 2021, doi: https://doi.org/10.5194/acp-21-13973-2021); NOy reactive uptake in clouds (Holmes et al., 2019, doi: https://doi.org/10.1029/2019GL081990); lightning-produced oxidants (Mao et al., 2021, doi: https://doi.org/10.1029/2021GL095740).

**Response:** This is a great suggestion and we have added these references to the analysis in Section 3.3.3 in Lines 572 - 588.

- The difference in correlations between GEOS-CF and GEOS-Chem (0.69 vs. 0.66) is not large enough to be of any statistical note (both round to 0.7). I suggest removing any discussion of this small difference in correlations. Statistically, the models perform the same.

**Response:** We agree with the reviewer's suggestion and have adjusted the manuscript to remove such discussions. Furthermore, Figure 6 has been moved to the Supplementary Materials as Figure S7 and the individual cluster correlations figure (previously Figure S4) has been moved to the main manuscript as Figure 7 to focus on more cluster-to-cluster differences. We thank the reviewer for their constructive comment as it improved the manuscript significantly.

**Minor comments:**

Line 57: "set out to address this issue" appears twice in the sentence. Please delete one.
**Response:** Thank you, this was a typo. We have corrected this.

Line 66-69: Ozonesondes are also able to resolve vertical levels. How does the vertical resolution of the lidar measurements compare to ozonesondes? What advantages do lidars have over sondes?
**Response:** We provided additional examples of the advantages of using the ozone lidar measurements and their value in understanding the full story behavior of ozone development throughout the manuscript. Please refer to Lines 520 – 537, Lines 568 – 571, and Lines 673 – 682 and in general the newly updated Section 3.3.3.

Line 264: "significantly" Please provide a p-value or level of significance.
**Response:** We are sorry for this confusion as we do not imply a statistical significance. We have removed 'significantly' to avoid further confusion.

Line 406: "modeled versus lidar observation spatial O3 differences" This is confusingly worded. The "differences" implies that you are performing a mathematical operation (model – observed), but what you are doing is plotting model vs observations in Figure 7. Please reword to better express that.
**Response:** Yes, we are providing the model minus the observed differences to visualize the performances of the models more clearly. I believe the confusion stems from the fact that Figure 6 and Figure 7 were accidentally switched in the submitted manuscript, but we have corrected this. We are sorry for the confusion.

See the correct Figure 7 below:

[Figure]

**Figure 7**. Spatial O$_3$ difference (model – lidar observations) for each cluster (1 – 5). GEOS-Chem differences (a) and GEOS-CF difference (b).

With the current changes to the manuscript this Figure 7 above is now permanently Figure 6. These changes were made to shift more focus on the different cluster specific model evaluation.

Line 589: Is the underestimate in wind speed and failure to reproduce wind shifts by GEOS-Chem explained by the use of offline meteorology? Some variables in MERRA-2 are averaged every 3 hours, and I assume that sea/bay breezes occur more rapidly than that.
**Response:** The reviewer makes a very good point. The meteorological variables are averaged every 3 hours which would influence the model's ability to simulate such fine scale temporal changes. Although GEOS-CF does run with online meteorology and does have a slight underestimation of winds as well. We have included this in the analysis.

Line 604-606: It is important to note that GEOS-Chem runs with offline meteorology, averaged every 3 hours. Since sea/bay breezes often happen at a finer temporal resolution, the GEOS-Chem model is at a disadvantage in modelling such fine processes.

Line 652 and 145: "Automatized" should be automated.
**Response:** Done.

**Figures:**
Figure 5. Can the observed winds be added to this plot, perhaps as a second row of panels?
**Response:** Done.

See the updated Figure 5 below:

[Figure]

**Figure 5.** Cluster averaged meteorological surface AQS station observations and GEOS-Chem model results. a) Surface temperature observations represented as the circular markers and simulated surface temperatures represented as the spatial contour (top-panel). b) Surface wind speed and direction observations represented as the circular markers and white arrows and simulated wind speed and direction represented as spatial contour and black arrows (bottom-panel).

Figure S1. Remove Table 2 title from the top of the table.
**Response:** Done. The figure has been fixed.

**References**

Holmes, C. D., Bertram, T. H., Confer, K. L., Graham, K. A., Ronan, A. C., Wirks, C. K., and Shah, V.: The Role of Clouds in the Tropospheric NO$x$ Cycle: A New Modeling Approach for Cloud Chemistry and Its Global Implications, Geophys. Res. Lett., 46, 4980–4990, https://doi.org/10.1029/2019GL081990, 2019.

Mao, J., Zhao, T., Keller, C. A., Wang, X., McFarland, P. J., Jenkins, J. M., and Brune, W. H.: Global Impact of Lightning- Produced Oxidants, Geophys. Res. Lett., 48, https://doi.org/10.1029/2021GL095740, 2021.

Wang, X., Jacob, D. J., Downs, W., Zhai, S., Zhu, L., Shah, V., Holmes, C. D., Sherwen, T., Alexander, B., Evans, M. J., Eastham, S. D., Neuman, J. A., Veres, P. R., Koenig, T. K., Volkamer, R., Huey, L. G., Bannan, T. J., Percival, C. J., Lee, B. H., and Thornton, J. A.: Global tropospheric halogen (Cl, Br, I) chemistry and its impact on oxidants, Atmospheric Chem. Phys., 21, 13973–13996, https://doi.org/10.5194/acp-21-13973-2021, 2021.

These references have been added to the manuscript. We thank the reviewer for their additions to the manuscript.

---

## Referee Report (RR1)

The manuscript titled "Cluster-based characterization of multi-dimensional tropospheric ozone variability in coastal regions: an analysis of lidar measurements and model results" by Claudia Bernier et al. developed a clustering analysis of multi-dimensional measurements of ozone in coastal regions. The lidar clusters provided a more comprehensive perspective to evaluate the performance of three-dimensional models. The manuscript provides valuable information for understanding ozone chemistry in complex coastal regions. I would recommend publication if my following comments are well addressed.

For the issue about the models' poor performance in simulating mid-level $O_3$, one influential process is the transport of $O_3$ in the free troposphere from the continent to coastal areas. I wonder whether your model can capture this process accurately.

Lines 513-530: You speculate that the overestimation of $O_3$ in the morning is because of underestimation of NO titration at night in the cluster MCO. Some evidence should be provided. The verification of modeled $NO_X$ by observed values can help to understand this issue. Also, in Fig. 6 it seems that GEOS-CHEM better captures the $O_3$ levels in the morning than GEOS-CF for the clusters MCO and HLO. The reason for the discrepancy between the two models' performance should be clarified.

Lines 531-532: The GEOS-CF model also overestimated the $O_3$ in the afternoon even if it does not overestimate early-morning $O_3$ (e.g., LLO and LMO). This means there exist processes other than nighttime NO titration causing the overestimation of $O_3$ in the afternoon. I suggest to point out this issue and try to explain the potential causes.

In Sec. 3.4.2: The specific effect of the models' performance simulating wind on $O_3$ simulation is not clearly explained. For example, how the wind leads to the overestimation of $O_3$ in MCO case by models, while leads to an underestimation of $O_3$ in HLO case by GEOS-CHEM. From my understanding, the wind will at least influence the dilution rate and horizontal and vertical transport of $O_3$.

Lines 623-635: I think this paragraph should focus on how the sea/bay breeze events cause a difference in $O_3$ profiles between and MCO and HLO cases and how they influence the model simulations. The reason why the two curtains are not in the same cluster is not important for your research objective as it is just an observed phenomenon.

The manuscript is too long and many sentences are redundant. There are a lot of repeated description in the main text, such as the description about the advantage of the lidar measurement and cluster approach, and the models' performance in low- and mid-levels. This will reduce the readability of the paper. I suggest to remove some redundant sentences. In addition, the conclusion is also too long. I suggest to simplify the conclusion and only convey the key information.

Other comments:

Line 335-336: "the HLO cluster reveals the specific case in which higher O3 is captured early in the temporal profile 336 in the low-level and translates to the higher O3 captured in the low-level as well". This sentence is ambiguous. Please rephrase it.

Lines 339-340: how do you infer that the cluster HMO indicates concentrated residual layers in the mid-level. Can you provide any evidence?

Lines 369-370: I suppose the low vertical mixing may reduce the descending $O_3$ from above level, leading to lower low-level $O_3$ concentrations.

Lines 370-371: "Relatively calm wind speeds and lower temperatures indicate other possible meteorological factors such as high cloud cover that could have contributed to the lower O3 concentrations in LLO". This sentence is unclear because lower temperature will also lead to lower $O_3$ production.

The title of Sec. 3.3.3 is not appropriate. You mainly discuss the potential causes that influence the model performance capturing the clusters' $O_3$ levels. A better title should be considered.

Line 511: Do you mean "despite having a low correlation in other cases"?

Line 526-527: "In HLO alone, there were 4 (out 527 of 18) of the profiles that were consecutive while in MCO there were 8 (out of 28)". This sentence is unclear. What do you mean by the word "consecutive".

Lines 536-547: The structure of this paragraph is weird. Since you mainly discuss the situation in low level, it is not appropriate to discuss the mid-level situation. It is better to move these sentences to the next paragraph where you mainly focus on the mid-level $O_3$.

Fig. 8. The symbol of wind direction is weird. Ordinary arrows are better to indicate wind direction. The arrows in panel (a) are too dense and the color is unclear. In addition, I didn't see the shift of wind direction from westerly to easterly winds in the MCO case. I suggest to define the meaning of arrow direction in the Figure legend.

Line 597: "led" should be "led to".

Lines 599-600: It seems that easterly winds prevail in the early morning and shift to northerly winds in the afternoon at the low level.

Line 630: I do not see that the MCO case has higher afternoon $O_3$ concentrations captured above 2000 m than the HLO case in Fig. 8d. They seem similar. In Fig. 4, the MCO cluster has lower afternoon $O_3$ concentrations captured above 2000 m than

the HLO cluster. In addition, does the white color in Fig.8d represent 0 ppb of $O_3$ or missing data?

Line 660: GEOS-CF has an overall lower unsystematic bias range.

---

## Referee Report (RR2)

**Major comment:**
The authors have made revisions that have improved the manuscript in response to the two reviewer comments. In addition to the specific comments below, the authors still need to clarify how the lidar data is better than ground, aircraft, or sonde data. There is no discussion in the abstract of why lidar data is needed, and in some of the discussion, it appears that surface data would suffice. I would support publication after the authors improve their discussion of how the ozone vertical structure information is beneficial over other data, and make that clear in the abstract and conclusions, and in their example discussions.

**Specific comments:**

Page 12, line 336 – The word 'background' in the ozone literature often refers to what ozone would be without anthropogenic influences. Consider using a different word to avoid confusion. The HLO curtain has lower free tropospheric ozone than the HMO profiles but higher surface ozone. So, it is likely that surface production is the most important factor here exacerbated by entrainment.

   Also, the following sentence is confusing. Did you mean to say "low-level" twice? "For example, the HLO cluster reveals the specific case in which higher $O_3$ is captured early in the temporal profile in the low-level and translates to the higher $O_3$ captured in the low-level as well."

Line 354 – Are the cluster average temperatures only at the LIDAR locations? Or across the whole domain?

Line 366 – Please remove 'slightly', if it is not a real difference don't discuss.

Line 380 – There is higher ozone at the surface in the MCO and HLO curtains in Figure 4. Why is MDA8 ozone highest in HMO? If you calculate MDA8 ozone from Figure 4, do your results agree? Or are the regulatory sites missing these elevated ozone concentrations observed by the lidar?

Figure 7 – Some of the mid-level correlations don't look statistically significant at all.

Line 439 – Are these numbers percentages? Is +0.30 actually +30%? Clarify that these are mean normalized biases.

Line 447 – You just told us that GEOS-Chem underpredicts high concentrations, but then say this challenges the assumption that models struggle to capture extreme cases. This seems to be a contradiction. It could help to quantify whether GEOS-Chem underpredicts at a certain percentage, say the 90th, or 95th etc percentile. You could also consider adding a probability distribution function to clarify your points.

Line 504 – This statement again would definitely benefit from a probability distribution plot.

Line 530 – The conclusion about multi-day events is a great one to consider including in the abstract as an example of how lidar data can help models. However, it needs an explanation about why surface ozone data would not be sufficient. The discussion seems to be mainly about the surface, so it is unclear why the vertical structure is needed here.

Line 572 – It seems that the best use for lidar would be in simulating elevated surface ozone that appears to be from transport and entrainment. I am surprised there isn't more discussion of this application.

Supplement.
Missing a '.' In the paragraph before Table S1 between 'needed' and 'These'.

---

## Author Response (AR3)

**Reply to Reviewers**

We thank the two reviewers for their constructive comments and suggestions, which have further improved the quality of our manuscript considerably. Their comments are reproduced below with our responses in blue. The corresponding edits in the manuscript are highlighted with tracked changes. The line numbers are based on the revised manuscript.

**Reviewer #1:**

The manuscript titled "Cluster-based characterization of multi-dimensional tropospheric ozone variability in coastal regions: an analysis of lidar measurements and model results" by Claudia Bernier et al. developed a clustering analysis of multi- dimensional measurements of ozone in coastal regions. The lidar clusters provided a more comprehensive perspective to evaluate the performance of three-dimensional models. The manuscript provides valuable information for understanding ozone chemistry in complex coastal regions. I would recommend publication if my following comments are well addressed.

For the issue about the models' poor performance in simulating mid-level O3, one influential process is the transport of O3 in the free troposphere from the continent to coastal areas. I wonder whether your model can capture this process accurately.

This is a great idea. To test this we used flight data from OWLETS-2 to evaluate the model's ability to simulate CO in the free troposphere (Figure S8). We evaluated just 6 flights (morning and evening flights) all which profiles belong to HMO, MCO, and HLO clusters. From this analysis we find that CO is simulated well in the free troposphere at lower levels (100 -110 ppbv) which is indicative of background levels of CO. On the other hand, the model handles higher levels (130 – 140 ppbv) of CO worse, with consistent underestimations in the free troposphere. Since higher levels of CO in the free troposphere are indicative of long-range transport from outside the study region, we can conclude that the GEOS-Chem model struggles to simulate transport in the free troposphere which directly influences the mid-level performance.

[Figure]

**Figure S8.** Aircraft measurements and GEOS-Chem simulated CO in the free troposphere during OWLETS-2. Measurements from the UMD Cessna 402B Research Aircraft.

Lines 517 – 523: "Additionally, transport of emissions in the free troposphere (FT) is another influential factor that could contribute to the misrepresentation of mid-level $O_3$. In Figure S8, aircraft measurements from OWLETS-2 are used to evaluate GEOS-Chem simulated CO in the FT (1800 – 2500 m AGL). The flight days evaluated are all curtain profiles that were assigned to the clusters with higher levels of $O_3$ in the mid-level (HMO, MCO, and HLO). It is evident that the model is able to capture lower levels of CO in the FT (100 – 110 ppbv) (e.g., background levels) but struggles to capture the higher levels (130 – 140 ppbv). Since increased levels of CO in the FT are indicative of possible long-range transport (Neuman et al., 2012), FT transport could be a factor contributing to the GEOS-Chem poor performance in the mid-level."

Lines 636 – 637: "Another factor inhibiting the poor simulation in the mid-level is the model failing to capture long-range transport of emissions in the FT."

Lines 513-530: You speculate that the overestimation of O3 in the morning is because of underestimation of NO titration at night in the cluster MCO. Some evidence should be provided. The verification of modeled NOX by observed values can help to understand this issue. Model overestimation of $O_3$ at night and in early morning hours is a common problem for 3-D Eulerian CTMs. It is thought to be caused by the model's inability to resolve the shallow surface layer at night, which enhances nighttime NO titration and $O_3$ dry deposition. Lines 481 – 485 have been revised.

Lines 481 – 485: "Model overestimation of $O_3$ at night and in early morning hours is a common problem for 3-D Eulerian CTMs. Overnight, $O_3$ concentrations from the evening before can remain lingering in the residual layer. This residual layer sits at about 1000 m or higher depending on the conditions of the environment. $O_3$ trapped in this residual layer can directly

correlate with the next day afternoon $O_3$ (e.g., Figure 3a; HLO cluster). Models struggle to resolve the shallow surface layer at night, which enhances nighttime NO titration and $O_3$ dry deposition. If this residual layer and the titration of $O_3$ overnight in the shallow surface layer is not resolved, next day simulated $O_3$ will most likely warrant even greater biases."

Also, in Fig. 6 it seems that GEOS-CHEM better captures the O3 levels in the morning than GEOS-CF for the clusters MCO and HLO. The reason for the discrepancy between the two models' performance should be clarified.
We have updated Lines 505 - 510 have been updated to emphasize this discrepancy.

Lines 505 - 510 : " GEOS-Chem does not have such an issue overestimating low-level $O_3$ in the afternoon. In the other clusters, GEOS-Chem actually underpredicts early morning low-level $O_3$ in the full vertical profile and does an overall better job than GEOS-CF simulating morning low-level $O_3$, such as in the HLO cluster. A better estimation of early morning $O_3$ does not warrant the same build-up of afternoon $O_3$. In these cases, GEOS-Chem handles the multi-day simulations better than GEOS-CF. This gives some explanation to why GEOS-Chem underpredicts the other clusters with higher $O_3$ concentrations in the low-level (HMO and HLO)."

Lines 531-532: The GEOS-CF model also overestimated the O3 in the afternoon even if it does not overestimate early-morning O3 (e.g., LLO and LMO). This means there exist processes other than nighttime NO titration causing the overestimation of O3 in the afternoon. I suggest to point out this issue and try to explain the potential causes.
This is an important distinction between the models and has been added to the manuscript (see Lines 528 – 532)

Lines 528 – 532: "GEOS-CF does best simulating morning low-level $O_3$ in cases of lower $O_3$ extent (LLO and LMO), but still overestimates the afternoon $O_3$. Since in these cases the afternoon does not seem to be related to morning estimations, other factors must be contributing. In the LLO cluster, the full curtain profile implies excessive mixing throughout the vertical profile is adding to afternoon $O_3$ overestimation. Similarly, for the LMO cluster, mid-level $O_3$ seems to be influencing low-level $O_3$ which could be adding to afternoon biases."

In Sec. 3.4.2: The specific effect of the models' performance simulating wind on O3 simulation is not clearly explained. For example, how the wind leads to the overestimation of O3 in MCO case by models, while leads to an underestimation of O3 in HLO case by GEOS-CHEM. From my understanding, the wind will at least influence the dilution rate and horizontal and vertical transport of O3.
Indeed, wind speeds are underestimated in all of the cases and by both models. This does help explain the overestimation of $O_3$ for both cases in GEOS-CF as well as the MCO case in GEOS-Chem but does not explain the underestimation in the HLO case for GEOS-Chem. The overestimation is easily explained as attributed by stagnant conditions in all the cases except for the HLO case by GEOS-Chem. The manuscript was updated to address and highlight the difference. One of the factors that could be contributing to the underestimation of $O_3$ in the HLO case could be the simulation of the boundary layer. We conclude the sea breeze mechanism is

not captured well by the GEOS-Chem model but better simulated by the GEOS-CF model likely due to resolution differences.

Lines 622 – 627: "Since the results reveal $O_3$ is underestimated, this suggests that there are more factors affecting $O_3$ results in this specific case. One of these factors can be the simulation of the boundary layer as the sea/bay breeze develops. If the boundary layer is simulated to be larger in depth, the ability for the model to simulate higher $O_3$ concentrations may be hindered such as found in Dacic et al. (2017). Since the HLO case indicates a common sea breeze event based on the timing and shift, it appears that GEOS-Chem really struggles capturing this intricate process while GEOS-CF does a better job."

Lines 623-635: I think this paragraph should focus on how the sea/bay breeze events cause a difference in O3 profiles between and MCO and HLO cases and how they influence the model simulations. The reason why the two curtains are not in the same cluster is not important for your research objective as it is just an observed phenomenon.
We agree with the reviewer and have added how the sea/bay breeze event can cause differences in the curtain profiles and how they influence the simulations (see Lines updated sect. 3.4.2 and Lines 628 – 637). But we have kept the final statement of why the curtains are not in the same cluster. We believe this is important since the purpose of categorizing the curtain profiles could suggest that two possible sea/bay breeze events should be in the same cluster. In highlighting the differences between the two cases, we reassure that the developed cluster method is still useful and actually proves the method's ability. We have updated Lines 633 – 634 to reflect this.

Lines 633 – 634: "Analyzing their full curtain profiles, it is easy to conclude why these events were not assigned to the same cluster and the differences are also apparent in the individual model performance."

Lines 628 – 637: "It is evident from these cases that differences in sea/bay breeze events can lead to diverse $O_3$ profiles. The HLO case has high $O_3$ levels that reach down to the surface, with peaks > 75 ppb at both 12:00 and again at 16:00 EDT. Just above this extreme $O_3$ plume at 2000 m, there is an $O_3$ deficit of almost 50 ppb. The MCO case differs in that the highest $O_3$ concentrations do not reach the surface. Also, $O_3$ is more distributed and mixed throughout the curtain profile and the vertical gradient, although present, is not as stark as the HLO case. The HLO cases also has higher $O_3$ captured aloft above 2500 m which is not captured in the MCO case. Analyzing their full curtain profiles, it is easy to conclude why these events were not assigned to the same cluster and the differences are also apparent in the individual model performance. For both cases, the models generally seem to underestimate wind speed and overestimate $O_3$ (to different extents) but the GEOS-Chem performance in the HLO case is different. The uniqueness of this case implies that GEOS-Chem struggles to simulate this sea/bay breeze based on factors other than wind speed and direction."

The manuscript is too long and many sentences are redundant. There are a lot of repeated description in the main text, such as the description about the advantage of the lidar measurement and cluster approach, and the models' performance in low- and mid-levels. This will reduce the readability of the paper. I suggest to remove some redundant sentences. In addition, the

conclusion is also too long. I suggest to simplify the conclusion and only convey the key information.

We agree with the reviewer's comment and have removed redundant sentences throughout the manuscript and simplified the conclusion (please see Section 4 for updates).

Other comments:

Line 335-336: "the HLO cluster reveals the specific case in which higher O3 is captured early in the temporal profile in the low-level and translates to the higher O3 captured in the low-level as well". This sentence is ambiguous. Please rephrase it.

This sentence has been updated for clarification.

Lines 335-226: "For example, the HLO cluster reveals a unique low-level case in which elevated (~ 1000 m) high $O_3$ concentrations are captured early in the temporal profile that translate to the higher $O_3$ concentrations at the surface later in the evening."

Lines 339-340: how do you infer that the cluster HMO indicates concentrated residual layers in the mid-level. Can you provide any evidence?

Evaluating individual profiles from HMO, there are some cases that show a residual layer aloft. These cases reveal layers of higher $O_3$ concentrations above 2000 m earlier in the temporal profile.

Here are some examples:

[Figure]

Due to lack of vertical velocity and vertical velocity variance data, we are not able to say whether or not these layers contribute to higher concentrations later in the profile near the surface as we state in Lines 342 – 344.

Lines 369-370: I suppose the low vertical mixing may reduce the descending O3 from above level, leading to lower low-level O3 concentrations.

Yes, this is true. We have added this to provide more clarification to the statement.

Lines 369-370: "…which can reduce any possible descending $O_3$ from aloft."

Lines 370-371: "Relatively calm wind speeds and lower temperatures indicate other possible meteorological factors such as high cloud cover that could have contributed to the lower O3

concentrations in LLO". This sentence is unclear because lower temperature will also lead to lower O3 production.

This sentence was reworded.

Lines 370-371: "Relatively calm wind speeds, lower temperatures, and other possible meteorological factors such as high cloud cover could have contributed to the lower $O_3$ concentrations in LLO."

The title of Sec. 3.3.3 is not appropriate. You mainly discuss the potential causes that influence the model performance capturing the clusters' O3 levels. A better title should be considered.

Section 3.3.3. Advantages of cluster approach and derived model conclusions

Line 511: Do you mean "despite having a low correlation in other cases"?

The overall correlation between the observations and the GEOS-CF for the mid-level is 0.22 (Figure S7) which is a poor correlation. When we calculate individual cluster correlations, the LMO cluster is revealed to have a good correlation. So, although the average mid-level correlation for GEOS-CF was low, there can still be some cases in which the GEOS-CF can reproduce $O_3$ pattern in the mid-level. This is an emphasis on how the use of a clustering approach to evaluate model performance rather than overall, summarized approach is beneficial in this case.

Line 511: "An even more profound case is exposed in which GEOS-CF has a strong correlation with mid-level $O_3$ in the LMO cases despite having a low correlation overall and in the other clusters."

Line 526-527: "In HLO alone, there were 4 (out 527 of 18) of the profiles that were consecutive while in MCO there were 8 (out of 28)". This sentence is unclear. What do you mean by the word "consecutive".

We used the consecutive to describe curtain profiles that were following one after the other in order, e.g., multi-day events. This was better clarified with a change to Line 527: "Given multiple cases of multi-day or consecutive high $O_3$ events from the lidar measurements…"

Lines 536-547: The structure of this paragraph is weird. Since you mainly discuss the situation in low level, it is not appropriate to discuss the mid-level situation. It is better to move these sentences to the next paragraph where you mainly focus on the mid-level O3.

We agree with the reviewer and have split the paragraph up. The first half of the original paragraph is now included in the paragraph before (Lines 527 – 533) and the second half of the original paragraph is now included in the paragraph after (Lines 534 – 544).

Fig. 8. The symbol of wind direction is weird. Ordinary arrows are better to indicate wind direction. The arrows in panel (a) are too dense and the color is unclear. In addition, I didn't see the shift of wind direction from westerly to easterly winds in the MCO case. I suggest to define the meaning of arrow direction in the Figure legend.

The wind symbols are not arrows but wind barbs. I have fixed the density of the barbs to make them clearer to see. The shift in the MCO cluster is slight and is earlier in the morning at about 06:00 EDT.

New Figure 8:

[Figure]

Line 597: "led" should be "led to".
Done.

Lines 599-600: It seems that easterly winds prevail in the early morning and shift to northerly winds in the afternoon at the low level.
The wind symbols are wind barbs which always depict where the winds are coming from. Therefore, for the HLO cluster the winds begin as northwesterly – westerly winds that briefly shift to south – southeasterly winds at about 11:00 EDT before shifting to south – southwesterly winds.

Figure 8. Profile curtains of wind speed/direction (a-c) and ozone (d-f) from the lidar (top panel), GEOS-Chem (middle panel), and GEOS-CF (bottom panel). Results from OWLETS-2 at HMI. Wind direction is depicted by wind barbs.

Line 630: I do not see that the MCO case has higher afternoon O3 concentrations captured above 2000 m than the HLO case in Fig. 8d. They seem similar. In Fig. 4, the MCO cluster has lower afternoon O3 concentrations captured above 2000 m than the HLO cluster.
This line has been removed. What we meant to highlight was the lower $O_3$ concentrations just at/above 2000 – 2500 m in the HLO cluster, not necessarily that the entire mid-level had lower $O_3$. As it is explained more thoroughly in the line after (Line 631) we have removed the Line 630 to avoid the confusion.

In addition, does the white color in Fig.8d represent 0 ppb of O3 or missing data?
The white in the Fig.8d is indeed missing data in the measured curtain profiles. The description for Fig.8 was edited to address this.

Figure 8. Profile curtains of wind speed/direction (a-c) and ozone (d-f) from the lidar (top panel), GEOS-Chem (middle panel), and GEOS-CF (bottom panel). Results from OWLETS-2 at HMI. Wind direction is depicted by wind barbs. The white spaces indicate missing data for both the a) wind and d) lidar curtain profiles.

Line 581: "The white spaces in both the wind and $O_3$ lidar indicate missing data."

Line 660: GEOS-CF has an overall lower unsystematic bias range.
Done.

**Reviewer #2:**
Major comment:
The authors have made revisions that have improved the manuscript in response to the two reviewer comments. In addition to the specific comments below, the authors still need to clarify how the lidar data is better than ground, aircraft, or sonde data. There is no discussion in the abstract of why lidar data is needed, and in some of the discussion, it appears that surface data would suffice. I would support publication after the authors improve their discussion of how the ozone vertical structure information is beneficial over other data, and make that clear in the abstract and conclusions, and in their example discussions.

Specific comments:

Page 12, line 336 – The word 'background' in the ozone literature often refers to what ozone would be without anthropogenic influences. Consider using a different word to avoid confusion. The HLO curtain has lower free tropospheric ozone than the HMO profiles but higher surface ozone. So, it is likely that surface production is the most important factor here exacerbated by entrainment.
The line has been changed to avoid confusion.
Line 336: The mean profile curtain indicates these cases did not have "clean air" to begin with which can allow a greater accumulation in the low-level in the afternoon.

Also, the following sentence is confusing. Did you mean to say "low-level" twice? "For example, the HLO cluster reveals the specific case in which higher O3 is captured early in the temporal profile in the low-level and translates to the higher O3 captured in the low-level as well."
This sentence has been updated for clarification.

Lines 335-226: "For example, the HLO cluster reveals a unique low-level case in which elevated (~ 1000 m) high $O_3$ concentrations are captured early in the temporal profile that translate to the higher $O_3$ concentrations at the surface later in the evening."

Line 354 – Are the cluster average temperatures only at the LIDAR locations? Or across the whole domain?

There were not stations exactly where the LIDARs were located so we used the stations nearest to the locations of the LIDAR placements to represent the meteorology at the surface during the campaign dates (shown in Figure 5). Line 351 is updated to specify this.

Line 351: "To support the lidar clustering results, daily averaged meteorological surface observations from AQS stations nearest to the lidar locations pertaining to the campaign period and GEOS-Chem surface model output were evaluated in regard to the five clusters."

Line 366 – Please remove 'slightly', if it is not a real difference don't discuss.
Done.

Line 380 – There is higher ozone at the surface in the MCO and HLO curtains in Figure 4. Why is MDA8 ozone highest in HMO? If you calculate MDA8 ozone from Figure 4, do your results agree? Or are the regulatory sites missing these elevated ozone concentrations observed by the lidar?
This is a very good observation. Indeed, the AQS stations seem to not capture the same elevated ozone that is observed by the lidar. Since the lidars were in some cases placed to capture "over water" concentrations, many (about half) of the higher $O_3$ curtain profiles assigned to the HLO cluster were from the lidar stationed "over water". These measurements were not replicated by the nearest AQS station.

Lines 381 – 382: "Since the AQS stations applied here were just the nearest station to the lidar placements, the MDA8 $O_3$ captured by the AQS stations do not necessarily reflect the high $O_3$ concentrations capture by the lidars."

Figure 7 – Some of the mid-level correlations don't look statistically significant at all.
We agree with this statement, hence why we emphasize how the models struggle to capture mid-level $O_3$ pattern. The cluster approach allows us to catch cases in which the models poorly simulate with extremely low correlations e.g., not statistically significant, but also cases where the model performance have statistical significance even though they may be low correlations.

Line 439 – Are these numbers percentages? Is +0.30 actually +30%? Clarify that these are mean normalized biases.
No, they are not percentages – they are calculated mean normalized biases. This is clarified throughout the manuscript and in the Supplementary Material.

Line 408 – 409: "The mean normalized biases for the five clusters displayed in Table S1 (in Supplementary Material)…"
Line 435 – 436: "…(subsequent cluster calculated normalized biases and correlation can be found in Table S1)."
Table S1. Calculated mean normalized bias and…

Line 447 – You just told us that GEOS-Chem underpredicts high concentrations, but then say this challenges the assumption that models struggle to capture extreme cases. This seems to be a contradiction. It could help to quantify whether GEOS-Chem underpredicts at a certain

percentage, say the 90th, or 95th etc. percentile. You could also consider adding a probability distribution function to clarify your points.

We stated in Line 447 that the GEOS-Chem model performs well in capturing extreme events in the low-level with "slight tendencies" to underpredicts high concentrations. The model even has the lowest biases in the cases compared to the rest of the clusters. Table S1 shows the mean normalized biases for these cluster range from R = 0.07 – 0.09. This are very low mean normalized biases. The model is not perfect but in comparison to the other clusters and the GEOS-CF model, the GEOS-Chem model is able to capture these extreme cases. This challenges the assumption that models struggle to capture extreme events.

Line 447: "These results suggest GEOS-Chem actually performs well in cases of high $O_3$ as well as cases of low $O_3$ with only a slight tendency to overpredict lower $O_3$ concentrations and underpredict higher $O_3$ concentrations."

Line 504 – This statement again would definitely benefit from a probability distribution plot.

We thank the reviewer for their suggestion, but we believe that the statistics provided in Table S1 support the statements made in this manuscript well. Also, the visual representation of the model – observation spatial $O_3$ difference in Figure 7 provides additional support for the conclusions derived.

Line 530 – The conclusion about multi-day events is a great one to consider including in the abstract as an example of how lidar data can help models. However, it needs an explanation about why surface ozone data would not be sufficient. The discussion seems to be mainly about the surface, so it is unclear why the vertical structure is needed here.

We have emphasized the advantage of using lidar curtain profiles is evaluating multi-day $O_3$ events.

Lines 513 – 516: "Overnight, $O_3$ concentrations from the evening before can remain lingering in the residual layer. This residual layer sits at about 1000 m or higher depending on the conditions of the environment. $O_3$ trapped in this residual layer can directly correlate with the next day afternoon $O_3$ (e.g., Figure 3a; HLO cluster). If this residual layer and the titration of $O_3$ overnight is not successfully simulated, next day simulated $O_3$ will most likely warrant even greater biases."

Lines 524 – 527: "Full vertical and temporal curtains provided by lidar instruments are essential in fully understanding the development and depletion of $O_3$ in these cases. The mean curtain profiles in Figure 3a indicate that what is captured at the surface (below 500 m) in the early morning does not represent what is captured in the residual layer (1000 m) by the lidar. Therefore, surface data would not be sufficient in evaluating a multi-day event."

Lines 664 – 666: "This feature is attributed to multi-day $O_3$ events where $O_3$ left in the residual layer can contribute to higher $O_3$ in the afternoon the next and proves to be a challenge for CTMs. Lidar curtain profiles prove to be essential in evaluating these multi-day cases as they can capture the full development and deposition of $O_3$ in the residual layer that is not provided in surface evaluations."

Line 572 – It seems that the best use for lidar would be in simulating elevated surface ozone that appears to be from transport and entrainment. I am surprised there isn't more discussion of this application.

Indeed, this is a great use of lidar measurements, and we have included a brief analysis on transport (Figure S8). We used flight data from OWLETS-2 to evaluate the model's ability to simulate CO in the free troposphere (Figure S8) to investigate transport. We evaluated just 6 flights (morning and evening flights) all which profiles belong to HMO, MCO, and HLO clusters. From this analysis we find that CO is simulated well in the free troposphere at lower levels (100 -110 ppbv) which is indicative of background levels of CO. On the other hand, the model handles higher levels (130 – 140 ppbv) of CO worse, with consistent underestimations in the free troposphere. Since higher levels of CO in the free troposphere are indicative of long-range transport from outside the study region, we can conclude that the GEOS-Chem model struggles to simulate transport in the free troposphere which directly influences the mid-level performance.

[Figure]

**Figure S8.** Aircraft measurements and GEOS-Chem simulated CO in the free troposphere during OWLETS-2. Measurements from the UMD Cessna 402B Research Aircraft.

Lines 517 – 523: "Additionally, transport of $O_3$ in the free troposphere (FT) is another influential factor that could contribute to the misrepresentation of mid-level $O_3$. In Figure S8, aircraft measurements from OWLETS-2 are used to evaluate GEOS-Chem simulated CO in the FT (1800 – 2500 m AGL). The flight days evaluated are all curtain profiles that were assigned to the clusters with higher levels of $O_3$ in the mid-level (HMO, MCO, and HLO). It is evident that the model is able to capture lower levels of CO in the FT (100 – 110 ppbv) but struggles to capture the higher levels (130 – 140 ppbv). Since increased levels of CO in the FT are indicative of possible long-range transport (Neuman et al., 2012), FT transport could be a factor contributing to the GEOS-Chem poor performance in the mid-level."

Lines 636 – 637: "Another factor inhibiting the poor simulation in the mid-level is the model failing to capture long-range transport of emissions in the FT."
Entrainment is briefly mentioned in the manuscript as advantages of lidar measurements, but we do not go into details of cases found in this study since we would need more data such as vertical velocity data and velocity data variance data to support this.

Supplement.
Missing a '.' In the paragraph before Table S1 between 'needed' and 'These'.
Done.